PREREGISTERED RESEARCH ARTICLE

# CBP/p300 activation promotes axon growth, sprouting, and synaptic plasticity in chronic experimental spinal cord injury with severe disability

**Franziska Müller**[1,º], **Francesco De Virgiliis**[1,º], **Guiping Kong**[1,º], **Luming Zhou**[1,º], **Elisabeth Serger**[1], **Jessica Chadwick**[1], **Alexandros Sanchez-Vassopoulos**[1], **Akash Kumar Singh**[2], **Muthusamy Eswaramoorthy**[3], **Tapas K. Kundu**[2,4], **Simone Di Giovanni**[1]*

**1** Department of Brain Sciences, Division of Neuroscience, Imperial College London, London, United Kingdom, **2** Transcription and Disease Laboratory, Molecular Biology and Genetics Unit, JNCASR, Bangalore, India, **3** Chemistry and Physics of Materials Unit, JNCASR, Bengaluru, India, **4** Neuroscience Unit, Jawaharlal Nehru Centre for Advanced Scientific Research, Bangalore, India

º These authors contributed equally to this work.

* s.di-giovanni@imperial.ac.uk

**Note:** As this is a Preregistered Research Article, the study design and methods were peer-reviewed before data collection. The time to acceptance includes the experimental time taken to perform the study. Learn more about Preregistered Research Articles.

## Abstract

The interruption of spinal circuitry following spinal cord injury (SCI) disrupts neural activity and is followed by a failure to mount an effective regenerative response resulting in permanent neurological disability. Functional recovery requires the enhancement of axonal and synaptic plasticity of spared as well as injured fibres, which need to sprout and/or regenerate to form new connections. Here, we have investigated whether the epigenetic stimulation of the regenerative gene expression program can overcome the current inability to promote neurological recovery in chronic SCI with severe disability. We delivered the CBP/p300 activator CSP-TTK21 or vehicle CSP weekly between week 12 and 22 following a transection model of SCI in mice housed in an enriched environment. Data analysis showed that CSP-TTK21 enhanced classical regenerative signalling in dorsal root ganglia sensory but not cortical motor neurons, stimulated motor and sensory axon growth, sprouting, and synaptic plasticity, but failed to promote neurological sensorimotor recovery. This work provides direct evidence that clinically suitable pharmacological CBP/p300 activation can promote the expression of regeneration-associated genes and axonal growth in a chronic SCI with severe neurological disability.

## Introduction

Spinal cord injury (SCI) is a devastating disease affecting millions of people worldwide. Severe SCI leads to permanent motor, sensory, and autonomic dysfunction that disrupts the quality of life of affected people. The management of severe SCI is nowadays limited to supportive care. Current physical rehabilitation has measurable but limited benefits after moderate SCI

**Data Availability Statement:** All relevant data are within the paper and its Supporting Information files.

**Funding:** ISRT translational award-P90397 to SDG Marina Romoli Onlus-P82836 to SDG Rosetrees Trust-P72986 to SDG Brain Research Trust-P73576 to SDG and J C Bose fellowship to TKK. The funders had no role in study design, data collection and analysis, decision to publish, or preparation of the manuscript.

**Competing interests:** The authors have declared that no competing interests exist.

**Abbreviations:** BMS, Basso Mouse Scale; CBP, CREB-binding protein; CNS, central nervous system; CSP, carbon nanosphere; CST, corticospinal tract; DRG, dorsal root ganglion; EE, enriched environment; HAT, histone acetyltransferase; PCAF, p300/CBP-associated factor; RAG, regeneration-associated gene; SCI, spinal cord injury; SH, standard housing; TOST, two one-sided tests.

but fails to improve recovery after more severe and chronic injuries. This permanent loss of function is primarily caused by disruption of the connectivity of long-distance and intraspinal axonal fibres that fail to regenerate and reconnect with the neural circuitry below the injury [1]. To date, the lack of axonal regeneration after injury has been attributed to 2 main interconnected factors: (i) the formation of a cellular inhibitory environment that promotes growth cone collapse and (ii) the lack of an intrinsic regenerative response [2]. Furthermore, abnormal activity in spinal circuits below the injury contributes to a progressive deterioration of sensorimotor function [3] that is likely intensified in patients with long-standing chronic SCI. Therefore, neuromodulation/rehabilitation and axonal regrowth strategies seek to promote activity-dependent neuroplasticity to improve sensorimotor recovery.

Accumulating evidence suggest that increasing neuronal activity not only contributes to axonal sprouting, but it also strengthens synaptic plasticity and stimulates targeted axonal regrowth, favouring reconnectivity and functional recovery [4,5]. Modulation of axonal plasticity and growth of both motor and sensory fiber tracts, including of sensory circuits below the spinal lesion, can be carried out by specific neurorehabilitation schemes to enhance recovery after SCI [5]. However, this increase in regenerative growth and sprouting only partially promotes functional recovery, and it remains insufficient for reconnectivity and reestablishment of function in severe spinal injuries.

Attempts to stimulate the axonal regenerative response within the injured central nervous system (CNS) have been only partially successful through the manipulation of independent transcription factors or cofactors, including c-JUN, pCREB, SMAD1, MYC, HDAC5, KLF4, KLF7, and STAT3 [6–15]. Accumulating evidence shows that epigenetic modifications can contribute to the transcription-dependent enhancement of the regeneration programme in sensory axons or the injured optic nerve [16–19]. Specifically, we found that the histone acetyltransferase (HAT) p300/CBP-associated factor (PCAF) acetylates the promoters of several regeneration-associated genes (RAGs) driving their expression after sciatic nerve injury and that PCAF overexpression promotes sensory axon regeneration across the injured spinal cord [18]. We and others have also shown that the HAT p300 can enhance optic nerve regeneration, promoting the expression of selected RAGs [17], while inhibiting class I histone deacetylases or HDAC3 partially promotes axonal regeneration of sensory axons following SCI [20]. However, modulation of these targets that were identified from injury-dependent paradigms have not translated into significant neurological recovery. We recently found changes in neuronal activity by housing mice in an enriched environment (EE) (large cages with toys, tunnels, running wheels, and enriched bedding) or following specific chemogenetic modulation of neural activity, induced epigenetic modifications, enabling active transcription, and regenerative growth. We next established that the CREB-binding protein (CBP) is the lysine acetyltransferase involved in this activity-dependent plasticity. Importantly, we showed that delivery of a small molecule specific activator of CBP/p300 named CSP-TTK21 promotes regenerative gene expression, axonal regeneration, plasticity, and functional sensorimotor improvement following acute SCI in rodents [21]. TTK21 is a HAT activator conjugated to a glucose-derived carbon nanosphere (CSP) able to cross the blood brain barrier, cell, and nuclear membranes with peak nuclear expression 3 days post-IP injection [22]. It was previously shown to promote neurogenesis and ameliorate memory deficits in tauopathy model in mice through increased histone acetylation and expression of genes involved in synaptic plasticity [22,23].

In addition, we have recently found that housing mice in an EE following a transection of the thoracic cord promotes significant sensorimotor recovery. These experiments suggest that housing animals in an EE might represent an "enriched" form of neurorehabilitation (S1 Fig). An alternative interpretation is that EE represents a more physiological environmental setting as opposed to standard housing (SH), which reflects an impoverished environment, especially

when these housing conditions are compared to patients, suggesting that EE should be used as "standard" housing condition.

The lack of integration between approaches aimed to promote regenerative molecular mechanisms with neurorehabilitation and neuromodulation after SCI remains a major limitation for repair in severe and chronic SCI, where reawakening a regenerative gene expression programme and stimulation of disrupted neural activity are especially challenging and potentially critical to repair and recovery. Additionally, mechanistic and therapeutic advances in chronic SCI with severe disability are especially rare and therefore represent a high priority.

## Hypothesis and relevance

Here, we hypothesize that the pharmacological stimulation of CBP/p300 activity will enhance regenerative gene expression during a growth refractory phase 12 weeks after spinal injury, while housing animals in an EE 1 week postinjury will stimulate neuronal activity, consolidate axonal and synaptic plasticity as opposed to animals housed in SH. Combined proteomics and transcriptomics studies indicate that EE activates physiological responses that are independent from CBP pathways as recently reported [21]. They include modulation of mitochondrial metabolism, calcium signalling, ion channels, axonal transport, and release of extracellular signalling vesicles among others, potentially allowing for synergistic benefits between the drug and EE.

Therefore, we postulate that CBP/p300 activator CSP-TTK21 on an EE housing baseline following the most clinically relevant chronic spinal cord injuries with severe disability might enhance neuronal plasticity, regeneration, and functional recovery, providing a better understanding of regenerative failure and filling a gap in the path to translation. Importantly, EE does not represent a specific form of focused rehabilitation, but rather a more physiological setting compared to SH that better reflects the human condition where patients are encouraged to engage in physical activities after a SCI. This explains the rationale for having animals in EE as baseline as opposed to having them in SH, which represents an impoverished artificial environment. Indeed, our study will test the effect of the CBP activator on this more "physiological" background in chronic SCI with severe disability.

The long-ranging implications of this work lie on testing a novel strategy based upon the CBP/p300 pharmacological activation to promote functional recovery after chronic SCI with severe disability, delivering the necessary preclinical evidence to support future clinical translation. Importantly, lack of validation of this hypothesis will also provide essential information allowing the scientific community to entertain alternative hypotheses. They include the possibility (i) that increases in regenerative gene expression and neural activity might not be synergistic in providing growth and plasticity; (ii) that recovery and repair might need task-specific neurorehabilitation; (iii) that reestablishment of functional circuitry needs targeting the spinal synapses; and finally (iv) that identification and manipulation of the extraneuronal wound healing and scarring processes in chronic SCI might need to be engaged along with favouring neuroplasticity.

## Experimental design

The experimental design asked the question of whether activation of CBP and the cognate protein p300 with the weekly systemic delivery of TTK21 would enhance gene expression that supports plasticity and regenerative growth after chronic spinal cord transection injury with severe disability. Since no data were available on TTK21 in chronic SCI, a transection model in mice has been chosen versus a contusion in rats because (i) it allows a more accurate anatomical definition of axonal regeneration and sprouting; (ii) it has much less variability

allowing for a more reliable statistical assessment keeping the number of animals relatively limited; and (iii) it needs a fraction of the drug required (in mice versus rats) that is produced in JNCASR, Bengaluru, India. Currently, the availability of the compound does not allow to initiate studies in rats in the short term; however, these studies will surely prompt further investigation in severe chronic contusion in rats, which is the next step towards human translation.

Specifically, adult 6- to 8-week-old C57Bl/6 mice received a spinal cord T9 transection injury that destroys the ascending sensory fibres in the dorsal columns bilaterally as well as most of the descending corticospinal, raphespinal, rubrospinal, and reticulospinal motor tracts, mimicking severe clinical SCI. This lesion leads to permanent severe impairment in sensorimotor function, severely limiting stepping up to at least 42 days postinjury, resembling clinical severe SCI (S1 Fig). All mice were placed in an EE 1 week after the SCI. The CBP activator TTK21 is coupled to slow-release carbon nanoparticles (CSP-TTK21) that allows a weekly administration maintaining stable activity levels in target tissues. Thus, CSP-TTK21 was administered via IP injections once per week starting 12 weeks after injury for 10 weeks. A control group of mice was treated with nanoparticles and vehicle alone (CSP). The CSP-TTK21 and control CSP nanoparticles were used at the dosage of 20 mg/kg that showed efficacy in subacute SCI as recently published [21].

Animals were killed at week 22 postinjury. Sprouting and regeneration of the dorsal columns were analyzed with the retrograde axonal tracer Dextran-488 (injected in the sciatic nerve in proximity of L4-L6 dorsal root ganglia (DRGs) 5 to 7 days before killing the animals). This allows visualization of ascending fibres in the dorsal columns into and across the injury site as previously shown in Di Giovanni's lab [21]. Sprouting and regeneration of the corticospinal tracts (CSTs) were analysed with stereotaxic injections of the neural tracer Dextran-tetramethylrhodamine and biotin (Dextran T&B), into the motor cortex 2 weeks before killing, which allows visualization of descending fibres into and across the injury site according to standard procedures in Di Giovanni's lab [24]. Axonal dieback as well as the number of fibres past the lesion site were normalized to the number of labelled fibres prior to the lesion as previously shown [24]. Given it closely correlates to locomotion, we also measured sprouting of serotoninergic raphe-spinal motor tracts with 5-HT immunohistochemistry as recently described [21]. To assess whether the CSP-TTK21 treatment enhances synaptic plasticity, we measured the number of inhibitory vGat or excitatory vGlut1 synaptic terminals in proximity of neuronal targets such as interneurons in the dorsal horns and motoneurons in the ventral horns of the spinal cord (ChAT or NeuN immunostaining) including in association with specific tracing of CST, sensory, or 5-HT fibres. This was carried out by fluorescent multilabelling experiments that were analyzed by confocal microscopy. Histone acetylation as read out of CBP/p300 activation was also evaluated in layer V neurons, raphe nuclei, and DRG neurons by immunofluorescence. The expression of several regeneration associated factors including ATF3, JUN, GAP43, SPRR1a, KLF7, pERK, and pSTAT3 was also studied by immunofluorescence in sensory and motor neurons.

In addition, we assessed locomotion, coordination, and sensorimotor integration by performing open field assessment with the Basso Mouse Scale (BMS) and the gridwalk tests, as recently shown [21]. Lastly, Von Frey test for mechanoception and mechanical allodynia as well as Hargreaves test for thermoception and thermal hyperalgesia were used to specifically assess the function of the ascending sensory tracts.

Please find a graphical summary (S2 Fig) and a summary table (Table 1) of the experimental design.

**Table 1. Table summarizing the experimental design and data analysis.**

| Research question | Hypothesis | Sampling plan | Statistical analysis | Outcomes that confirm or disconfirm hypothesis |
|---|---|---|---|---|
| Does weekly systematic CSP-TTK21 delivery in a chronic severe SCI in mice housed in EE promote *sensorimotor fiber regeneration*? | We hypothesize that treatment of CSP-TTK21 in mice housed in EE will enhance sensorimotor fiber regeneration due to the synergy between the TTK21 and EE enabling fibres to regenerate through the site of injury and provide directional regrowth. Since EE is believed to enhance activity in both descending motor axons and facilitate motor control through the recruitment of proprioceptive feedback circuits, which are the pathways targeted by CBP/p300 activator, we expect a strong synergy between the drug and EE. | TTK21 vs. control: increase in % of axons regenerating beyond the lesion site from 5 to 10; SD = 2 and 4; Power 90%; P: 0.05, N: 8 | For 5-HT immunostaining and for quantification of axonal regeneration of CST and dorsal column fibres: Normally distributed data will be evaluated using a two-way repeated measures ANOVA with a Greenhouse-Geisser correction and a Tukey or Sidak post hoc test will be applied to examine multiple comparisons using a 95% confidence interval. For nonparametric evaluation, a Brunner & Langer nonparametric longitudinal data model could be used. For VGlut and VGat quantification, an unpaired two-tailed Student $t$ test with Welch correction and a 95% confidence interval will be applied. For nonparametric evaluation, a Mann–Whitney test will be used. A threshold level of significance α was set at $P$ value <0.05. Significance levels will be defined as follows: * $P$ value <0.05; ** $P$ value <0.01; *** $P$ value <0.001, **** $P$ value <0.0001. All data analysis will be performed blind to the experimental group. | Hypothesis would be confirmed if significant regeneration of sensory and motor (CST, raphespinal axons) and 5HT fibres are observed following CSP-TTK21 treatment compared to control. **CONFIRMED** Additional evidence to support this hypothesis would be provided by showing enhanced synaptic plasticity via a significant increase in the number of inhibitory VGat or excitatory VGlut synaptic terminals in proximity of neuronal targets in the spinal cord. **CONFIRMED for VGlut NOT CONFIRMED for VGat** |
| Does CSP-TTK21 systemic weekly delivery in mice housed in EE following a chronic and severe SCI enhance *functional recovery*? | We hypothesize that treatment of CSP-TTK21 in mice housed in EE will promote functional recovery. | Gridwalk: Control vs. TTK21: changes in score from 4 to 2; SD: 2 and 1; Power 90%; P: 0.05; N = 12 | Gridwalk and BMS: Normally distributed data will be evaluated using a two-way repeated measures ANOVA and a Tukey or Sidak post hoc test will be applied to examine multiple comparisons using a 95% confidence interval. For nonparametric evaluation, a Brunner & Langer nonparametric longitudinal data model could be used. A threshold level of significance α was set at $P$ value <0.05. Significance levels will be defined as follows: * $P$ value <0.05; ** $P$ value <0.01; *** $P$ value <0.001, **** $P$ value <0.0001. All data analysis will be performed blind to the experimental group. | Hypothesis would be confirmed if significant differences were observed in the BMS and gridwalk scores between CSP-TTK21 and control. We would expect scores to remain similar until CSP-TTK21 injection, where we expect the CSP-TTK21 group to show an improvement in scores while no improvement is seen in the SP-Veh control. **NOT CONFIRMED** |
| Does CSP-TTK21 systemic weekly delivery in mice housed in EE following a chronic and severe SCI promote *gene expression and histone acetylation* to enhance plasticity and regenerative growth? | We hypothesize that stimulating CBP/p300 activity will enhance regenerative gene expression and histone acetylation during a refractory phase 12 weeks postinjury, while placing mice into EE 1 week postinjury stimulates neuronal activity and consolidates axonal and synaptic plasticity. | TTK21 vs. control: change in fluorescence intensity of 1.5-fold; (between 50 and 75 arbitrary units, for example); SD = 10 and 12; P: 0.05; Power 90%; N = 4 | IHC/IF: Normally distributed data will be evaluated using a two-tailed unpaired Student $t$ test with Welch correction or a one-way ANOVA with a 95% confidence interval when experiments contained more than 2 groups. The Tukey post hoc test will be applied when appropriate. The Mann–Whitney $U$ test will be used for nonparametric evaluation. A threshold level of significance α was set at $P$ value <0.05. Significance levels will be defined as follows: * $P$ value <0.05; ** $P$ value <0.01; *** $P$ value <0.001, **** $P$ value <0.0001. All data analysis will be performed blind to the experimental group. | Hypothesis would be confirmed if a significant increase in gene expression of regeneration associated genes (e.g. c-Jun, Atf3, pStat3) as well as increased histone acetylation (e.g. H3k27ac, H3k9ac, H4k8ac) was observed following CSP-TTK21 treatment versus control. **CONFIRMED for selected genes** |

BMS, Basso Mouse Scale; CST, corticospinal tract; EE, enriched environment; IF, immunofluorescence; IHC, immunohistochemistry; SCI, spinal cord injury.

## Results

### CBP/p300 activator TTK21 promotes histone acetylation and the expression of regeneration-associated signals in neurons in chronic SCI

Delivery of the CBP/p300 activator TTK21 promoted histone acetylation as expected in DRG (Fig 1A–1D), raphe (Fig 1E and 1F), and layer V cortical neurons (Fig 1G–1J) at 22 weeks after SCI. Further, as shown by immunofluorescence experiments, TTK21 enhanced the expression of ATF3, SPRR1a, cJUN, KLF7, and GAP43 (Fig 2A–2J), however not of pERK and pSTAT3 (Fig 2K–2N) in DRG neurons. ATF3, KLF7, pSTAT3, and pERK were not expressed above background in layer V cortical neurons, while cJUN and SPRR1a were unaffected by TTK21 (S3 Fig). Thus, CBP/p300 activation promotes histone acetylation in both motor and sensory neurons as well as the expression of RAGs in sensory DRG neurons in chronic SCI.

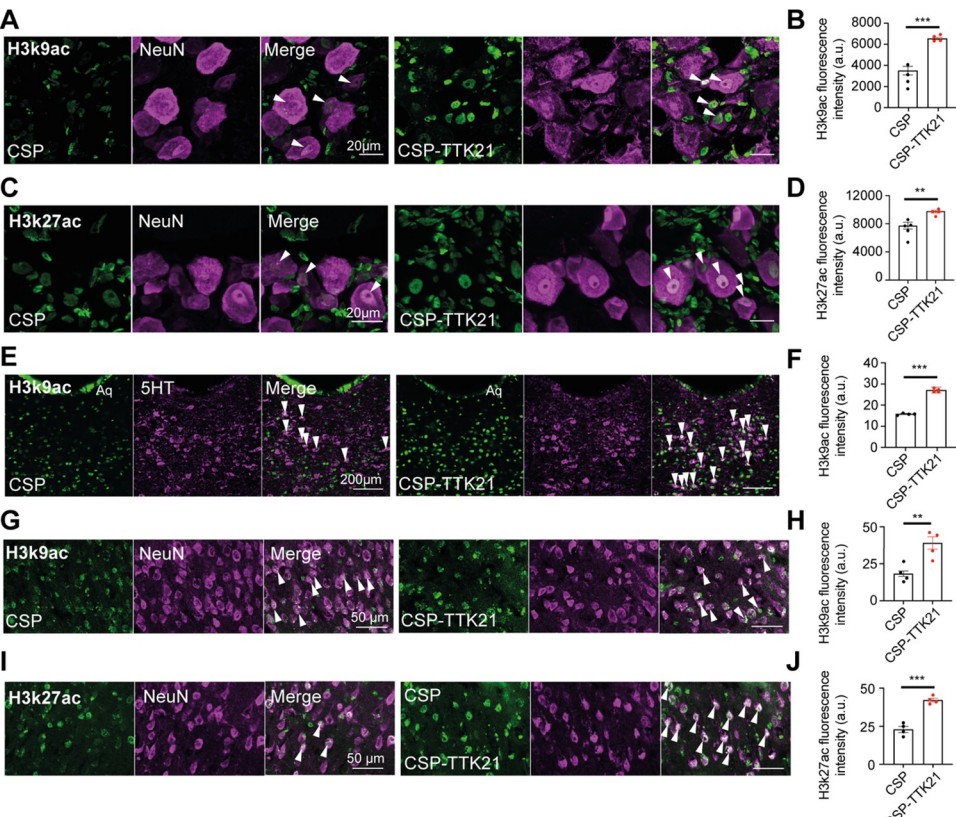

**Fig 1. Histone acetylation in DRG, raphe, and layer 5 cortical neurons in CSP-TTK21-treated mice in a chronic SCI with severe disability. (A)** Representative micrographs of H3K9ac immunostaining (green, white arrows) in DRG neurons. (**B**) Quantification of H3K9ac immunostaining in DRG neurons from CSP or CSP-TTK21-treated mice (CSP: 3,505.0 ± 399.8; CSP-TTK21: 6,549.0 ± 122.1, $p < 0.001$, $n = 4$). (**C**) Representative micrographs of H3K27ac immunostaining (green, white arrows) in DRG neurons. (**D**) Quantification of H3K27ac immunostaining in DRG neurons in CSP or CSP-TTK21-treated mice (CSP: 7,738.0 ± 472.0, $n = 5$; CSP-TTK21: 9,779.0 ± 195.4, $p < 0.01$, $n = 4$). (**E**) Representative micrographs of H3K9ac staining (green, white arrows) in raphe neurons. (**F**) Quantification of H3K27ac immunostaining in raphe neurons from CSP or CSP-TTK21-treated mice (CSP: 15.9 ± 0.2; CSP-TTK21: 27.1 ± 1.4, $p < 0.001$, $n = 4$). (**G**) Representative micrographs of H3K9ac immunostaining (green, white arrows) in layer 5 cortical neurons. (**H**) Quantification of H3K9ac immunostaining in layer 5 cortical neurons from CSP or CSP-TTK21-treated mice (CSP: 18.2 ± 1.8; CSP-TTK21: 39.0 ± 4.1, $p < 0.01$, $n = 4$). (**I**) Representative micrographs of H3K27ac staining (green, white arrows) in layer 5 cortical neurons. (**J**) Quantification of H3K27ac immunostaining in layer 5 cortical neurons from CSP or CSP-TTK21-treated mice (CSP: 22.9 ± 2.9; CSP-TTK21: 42.1 ± 1.3, $p < 0.001$, $n = 4$). Mean ± SEM; unpaired two-tailed Student $t$ test; ** $p < 0.01$, *** $p < 0.001$. $n$ = biologically independent animals. The data can be found in S1 Data. CSP, carbon nanosphere; DRG, dorsal root ganglion; SCI, spinal cord injury.

## CBP/p300 activator TTK21 enables axonal growth and synaptic plasticity in chronic SCI

Delivery of the CBP/p300 activator TTK21 significantly reduced axonal retraction and promoted axonal growth of motor corticospinal (Fig 3A and 2B) and of sensory DRG neurons, respectively (Fig 3C and 3D). However, the most striking phenotype was the increased sprouting of 5-HT raphe-spinal axons as shown by 5-HT immunofluorescence (Fig 3E and 3F). Importantly, TTK21 led to an increased number of vGlut1, but not of vGat, boutons opposed to motor neurons in the ventral horn of L1-3 spinal sections below the injury site (Fig 3G–3L).

No difference in the size of the lesion and in GFAP astrocytic immunostaining was observed between the experimental groups (S4A–S4C Fig). Similarly, measurement of macrophage/microglia CD68 immunofluorescence intensity around the injury did not show any difference between vehicle and TTK21 (S5 Fig), suggesting that systemic activation of CBP/p300 does not lead to obvious changes in the spinal cord astrocytic and macrophage/microglia responses. Thus, CBP/p300 activation can promote axonal growth and synaptic plasticity in chronic SCI with severe disability.

## CBP/p300 activator TTK21 does not affect neurological recovery in chronic SCI

Lastly, we measured whether CBP/p300 activation promotes sensorimotor recovery in chronic SCI. We did not observe any significant difference in the sensorimotor performance by BMS (Fig 4A) and gridwalk (Fig 4B) between TTK21 and vehicle-treated animals, while the severity of the lesion was confirmed by the severe impairment observed (Fig 4C). Only a minority of animals were able to perform stepping on a gridwalk (BMS >3; Fig 4B). Similarly, the animals that could be tested for mechanoception by Von Frey and thermal responses by Hargreaves showed severe impairments with no difference between the 2 experimental groups (Fig 4D and 4E). Thus, CBP/p300 activation cannot stimulate sensory or motor recovery with this level of chronic SCI.

## Discussion

Chronic severe SCI that is associated with absent axonal regrowth and reconnectivity and severe neurological disability remains a major medical challenge. It is believed that the ability for axonal growth and sprouting declines over time after an SCI and that the time window for intervention might eventually close. The importance of our findings lies on the evidence that starting a treatment such as the CBP/p300 activator TTK21 12 weeks after an SCI in the mouse with severe disability can still elicit a regenerative response as shown by increased axonal growth, sprouting, and synaptic plasticity at 22 weeks postinjury. The CBP/p300 activator TTK21 also promotes the expression of regenerative signals including some well-established RAGs. This is a potentially exciting discovery since it provides a demonstration that a clinically suitable molecular intervention can promote plasticity and growth in both an acute, as previously shown [21], and chronic SCI with severe disability, likely by reawakening a dormant regenerative gene expression programme. While both motor and sensory neurons showed increased histone acetylation, CBP/p300 activation induced the expression of most RAGs investigated in DRG neurons only. These findings are in line with previous work in subacute SCI where these classical RAGs were not activated in the corticospinal [25], which temporarily revert to an embryonic state, as opposed to DRG neurons, which reexpress RAGs, as shown here. However, the increase in histone acetylation seems to be a common denominator of TTK21 delivery for sensory and motor neurons alike. Increased acetylation likely stimulates

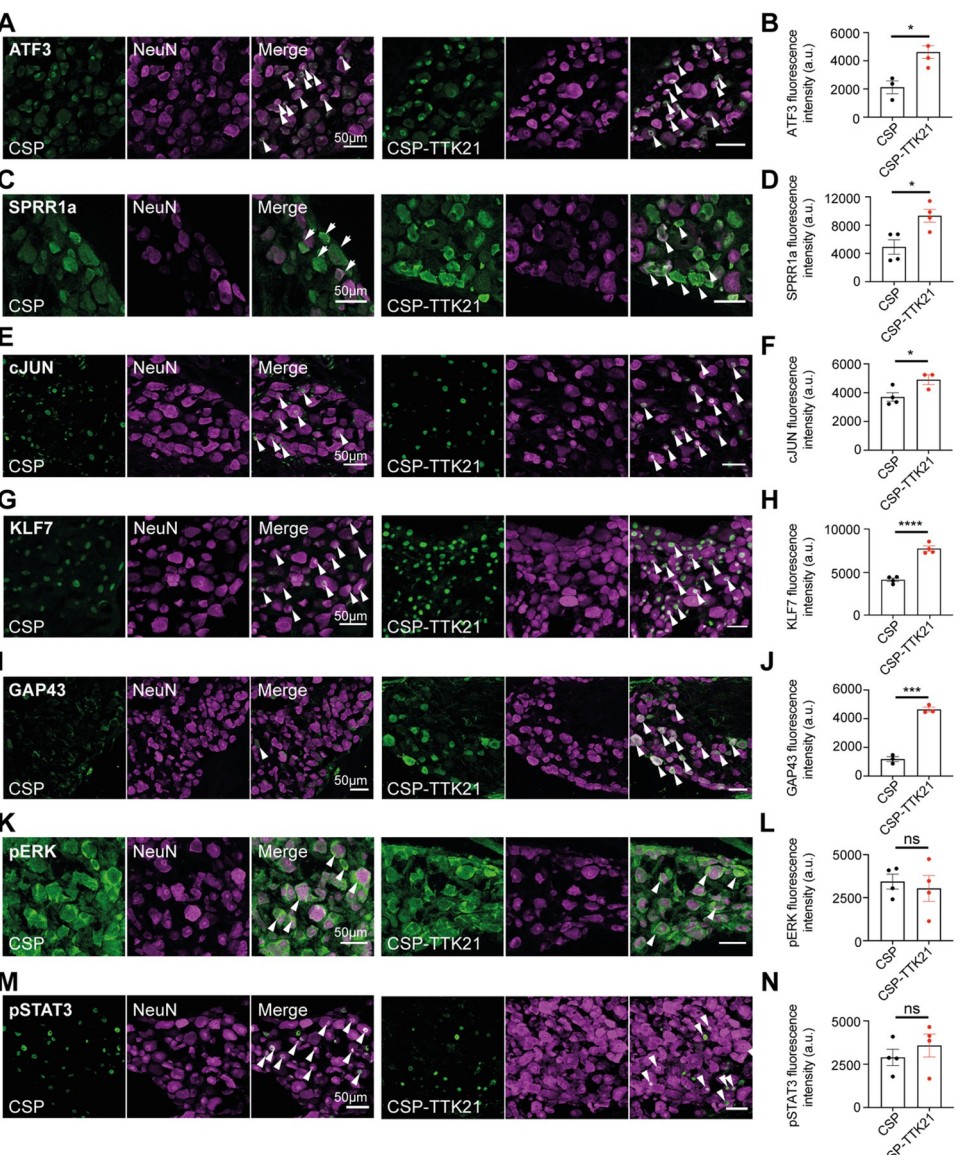

**Fig 2. Expression of regeneration-associated proteins in DRG neurons in CSP-TTK21-treated mice in a chronic SCI. (A-N)** Representative micrographs of immunostaining for regeneration-associated proteins (green, white arrows) in DRG neurons from CSP or CSP-TTK21-treated mice (A, C, E, G, I, K, M) and respective quantification of immunostaining in DRG neurons (B, D, F, H, J, L, N). **(A, B)** Atf3 (CSP: 2,113.0 ± 461.5; CSP-TK21: 4,614.0 ± 449.4, $p < 0.05$). **(C, D)** Sprr1a (CSP: 4,904.0 ± 1,033.0; CSP-TK21: 9,331.0 ± 914.6, $p < 0.05$). **(E, F)** cJun (CSP: 3,697.0 ± 302.2; CSP-TK21: 4,904.0 ± 340.7, $p < 0.05$). **(G, H)** Klf7 (CSP: 4,097.0 ± 199.1; CSP-TK21: 772.00 ± 317.9, $p < 0.0001$). **(I, J)** Gap43 (CSP: 1,178.0 ± 173.2; CSP-TK21: 4,643.0 ± 163.6, $p < 0.001$). **(K, L)** pErk (CSP: 3,433.0 ± 428.9; CSP-TK21: 3,036.0 ± 750.7, $p = 0.66$, TOST: t(4.8) = 0.03, $p = 0.51$ given equivalence bounds of −366.7 and 366.7 on a raw scale and alpha of 0.05). **(M, N)** pStat3 (CSP: 2,894.0 ± 470.1; CSP-TK21: 3,579.0 ± 656.8, $p = 0.43$, TOST: t(5.4) = −0.42, $p = 0.66$ given equivalence bounds of −342.7 and 342.7 on a raw scale and alpha of 0.05). Mean ± SEM; unpaired two-tailed Student or Welch $t$ test; $^{*}p < 0.05$, $^{***}p < 0.001$, $^{****}p < 0.0001$. $n = 3$ or 4 biologically independent animals. The data can be found in S1 Data. CSP, carbon nanosphere; DRG, dorsal root ganglion; SCI, spinal cord injury; TOST, two one-sided tests.

distinct regenerative programmes between peripheral sensory and central motor neurons by enhancing chromatin accessibility at gene regulatory regions of RAGs for sensory neurons and perhaps of developmentally regulated genes for corticospinal neurons as recent work from the Tuszynski's lab might suggest [25]. They found that reexpression of embryonic genes after SCI

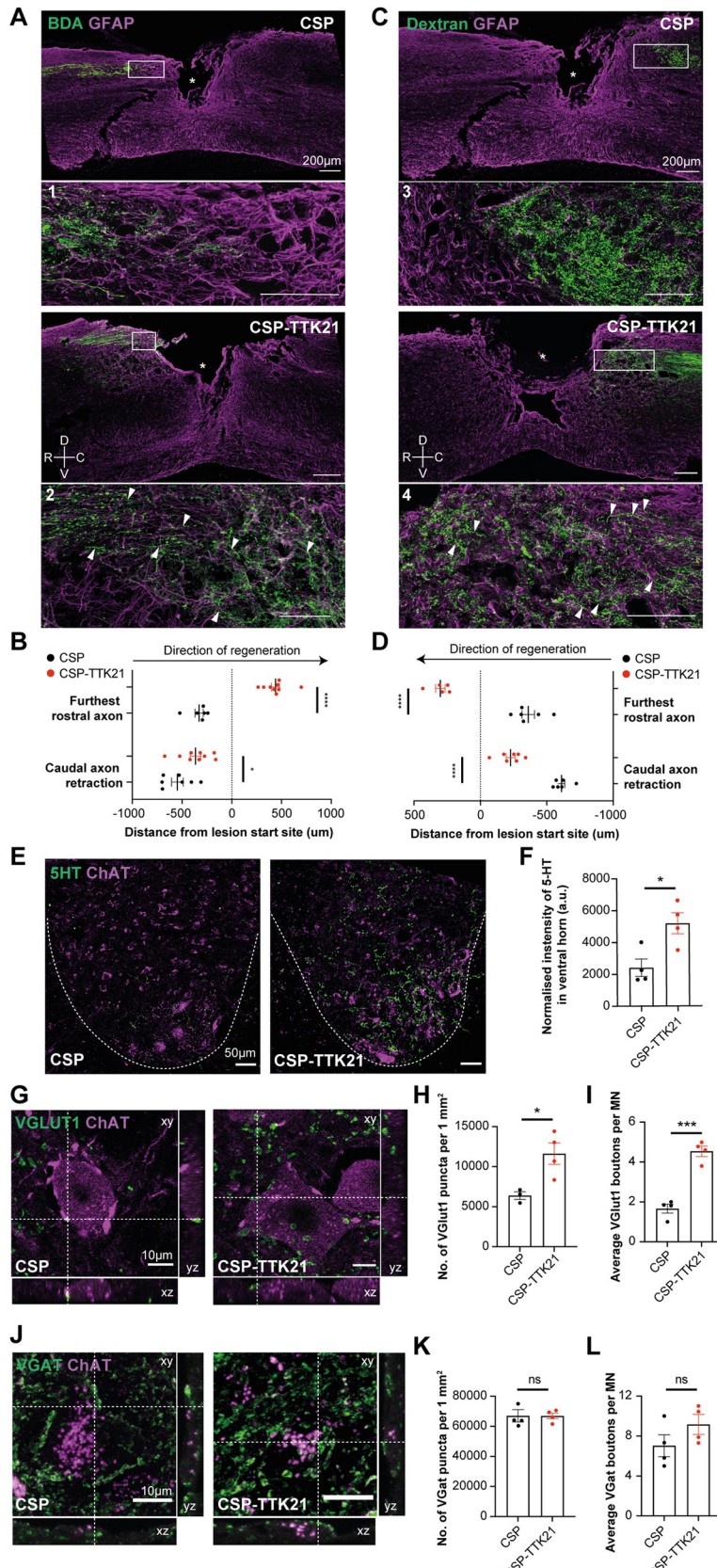

**Fig 3. Axonal growth and synaptic plasticity in CSP-TTK21-treated mice in a chronic SCI.** (**A**) Representative micrographs of BDA-traced CST axons (green, white arrows) after chronic SCI in CSP or CSP-TTK21-treated mice, GFAP (magenta) was used to determine the lesion site. (**B**) Average distance between the lesion border and the furthest rostral (CSP: −329.8 ± 41.4, $n = 6$; CSP-TTK21: 438.3 ± 40.4, $p < 0.0001$, $n = 9$) or caudal axons (CSP: −548.6 ± 58.7, $n = 7$; CSP-TTK21: −366.2 ± 54.6, $p < 0.05$, $n = 9$) in CSP and CSP-TTK21-treated mice. (**C**) Representative micrographs of Dextran-traced sensory axons (green, white arrows) after chronic SCI in CSP or CSP-TTK21-treated mice. GFAP (magenta) was used to determine the lesion site. (**D**) Average distance between the lesion border and furthest rostral (CSP: −361.2 ± 47.3, $n = 6$; CSP-TTK21: 301.6 ± 35.8, $p < 0.0001$, $n = 5$) or caudal axons (CSP: −610.5 ± 25.2, $n = 6$; CSP-TTK21: −227.8 ± 22.7, $p < 0.0001$, $n = 7$) in CSP and CSP-TTK21-treated mice. (**E**) Representative micrographs of 5-HT-positive axons (green) sprouting around ChAT-positive motoneurons (magenta) in the lumbar ventral horn (white dotted lines) below the lesion site in CSP or CSP-TTK21-treated mice. (**F**) Quantification of 5-HT intensity in the lumbar ventral horn (CSP: 2,410.0 ± 549.3; CSP-TTK21: 5,189.0 ± 662.2, $p < 0.05$, $n = 4$). (**G**) Orthogonal 3D confocal images of VGlut1+ boutons (green) from group-1a afferents in proximity to motoneurons (ChAT+, magenta) in the lumbar ventral horn below the lesion site. Intersection of dotted lines indicates example of apposition. (**H**) Quantification of Vglut1+ puncta per 1 mm$^2$ in CSP or CSP-TTK21 mice (CSP: 6,380.0 ± 483.1, $n = 3$, CSP-TTK21: 11,615.0 ± 1,337.0, $n = 4$, $p < 0.05$). (**I**) Quantification of VGlut1+ boutons in proximity to motoneurons (CSP: 1.7 ± 0.2; CSP-TTK21: 4.5 ± 0.3, $p < 0.0001$, $n = 4$). (**J**) Orthogonal 3D confocal images of VGat+ boutons (green) in proximity to motoneurons (ChAT+, magenta) in the lumbar ventral horn below the lesion site. Intersection of dotted lines indicates example of apposition. (**K**) Quantification of VGat+ puncta per 1 mm$^2$ (CSP: 66,366.0 ± 2,970.0; CSP-TTK21: 66,996.0 ± 2,069.0, $p = 0.87$, TOST: t(5.4) = 0.2, $p = 0.41$ given equivalence bounds of −1,535.7 and 1,535.7 on a raw scale and alpha of 0.05, $n = 4$). (**L**) Quantification of VGlut1+ boutons in proximity to motoneurons (CSP: 7.0 ± 1.1; CSP-TTK21: 9.3 ± 0.9, $p = 0.15$, TOST: t(5.7) = −1.2, $p = 0.86$ given equivalence bounds of −0.6 and 0.6 on a raw scale and alpha of 0.05, $n = 4$). All data are given as mean ± SEM; unpaired two-tailed Student t test; * $p < 0.05$, *** $p < 0.001$, **** $p < 0.0001$. $n$ = biologically independent animals. The data can be found in S1 Data. CSP, carbon nanosphere; CST, corticospinal tract; SCI, spinal cord injury; TOST, two one-sided tests.

lasts for 2 weeks only in corticospinal neurons. In the present chronic state of injury (12 to 22 weeks after injury), this "primed" state for regeneration has likely closed. Thus, delivery of CSP-TTK21 might have partially reopened this regenerative window. It is in fact important that TTK21 increases the growth and regenerative gene expression ability of both sensory and motor neurons, albeit with differential potency and efficacy, being maximal for regenerative gene expression at selected RAGs in sensory DRG neurons and for axonal sprouting in 5-HT motor neurons. The very distinct embryonic origin and molecular identities of these CNS neuronal subpopulations might underline this differential response [26]. However, the variable distance from the lesion site and the rate of axonal transport of selected neuronal populations might be additional contributing factors since neuronal cell types and the distance of the neuronal cell body from the injury site affect the rate and extent of axonal trafficking and the expression of RAGs.

Given the severity of the spinal injury in the present study, it is very difficult to directly compare these findings with the level of axonal growth, sprouting, and synaptic plasticity observed in our previous work when we delivered TTK21 6 or 24 hours after a mouse spinal cord hemisection or a rat moderate to severe contusion injury, respectively [21]. However, the most striking difference between the present and the previous experiments is the lack of any neurological recovery. The most immediate explanation might be the modest effect of axonal growth of sensory and corticospinal neurons, the depth of the transection injury in the present study compared to a dorsal hemisection or a moderate to severe contusion in the previous. The presence of astrocytic rich tissue bridges is important to allow for axonal growth, and the present chronic injury showed poor tissue bridges. Additionally, the diverse injury severities and timing postinjury likely reflect a different glial environment that might influence synaptic transmission, repair, and functional recovery. Finally, in a chronic condition, synaptic transmission might be compromised by the functional impairment of neuronal targets below (motor targets) or above (sensory targets) the injury site due to the long-standing deafferentation.

However, our positive neuroanatomical and molecular findings despite the limitations of the lack of neurological recover might pave the way for the future combinatorial use of TTK21

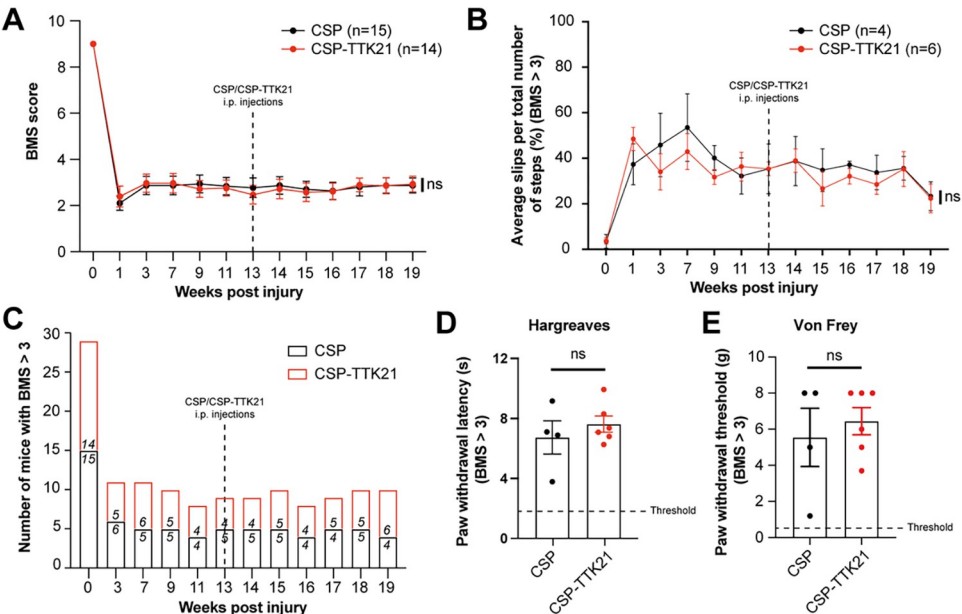

**Fig 4. Sensorimotor behavioural tests after CSP-TTK21 treatment in chronic SCI with severe disability. (A)** BMS quantification of CSP ($n = 15$) or CSP-TTK21 ($n = 14$) treated mice after chronic SCI (Treatment: f(1) = 0.002, $p = 0.96$ (TOST: given equivalence bounds of −0.21 and 0.21, 90% confidence intervals fall −0.37 and 0.32 and $p = 0.307$, thus H0 is undecided); Time: f(12) = 192.1, $p < 0.001$ (TOST: given equivalence bounds of −0.21 and 0.21, 90% confidence intervals fall −0.24 and −0.15 and $p = 0.215$, thus H0 is rejected); Interaction (treatment × time): f(12) = 6.6, $p = 0.82$ (TOST: given equivalence bounds of −0.21 and 0.21, 90% confidence intervals fall 4.19 and 4.99 and $p > 0.999$, thus H0 is rejected). Two-way repeated measures ANOVA with Sidak post hoc test. **(B)** Gridwalk quantification of the percentage of slips per total number of steps per run in CSP ($n = 4$) or CSP-TTK21 ($n = 6$) treated mice with a BMS score greater than 3 after chronic SCI (Treatment: f(1) = 0.2, $p = 0.69$ (TOST: given equivalence bounds of −1.78 and 1.78, 90% confidence intervals fall −7.96 and 2.63 and $p = 0.693$, thus H0 is undecided); Time: f(12) = 7.1, $p < 0.001$ (TOST: given equivalence bounds of −1.78 and 1.78, 90% confidence intervals fall −0.73 and 0.66 and $p < 0.001$, thus H0 is accepted); Interaction (treatment × time): f(12) = 6.6, $p = 0.82$ (TOST: given equivalence bounds of −1.78 and 1.78, 90% confidence intervals fall −28.59 and 41.29 and $p > 0.999$, thus H0 is rejected). Two-way repeated measures ANOVA with Sidak post hoc test). **(C)** Bar graph indicating the number of mice with a BMS score greater than 3 across postinjury time points. **(D)** Hargreaves test indicating average paw withdrawal latency in CSP or CSP-TTK21 (Welch two-tailed $t$ test: CSP: 6.7 ± 1.1, $n = 4$; CSP-TTK21: 7.6 ± 0.5, $n = 6$, $p = 0.51$, TOST: t(8) = −0.3, $p = 0.63$ given equivalence bounds of −0.5 and 0.5 on a raw scale and an alpha of 0.05) treated mice with a BMS score greater than 3. **(E)** Von Frey test indicating average paw withdrawal threshold in CSP or CSP-TTK21 (Welch two-tailed $t$ test: CSP: 5.5 ± 1.6, $n = 4$; CSP-TTK21: 6.4 ± 0.7, $n = 6$, $p = 0.64$, TOST: t(4.3) = −0.1, $p = 0.52$ given equivalence bounds of −0.8 and 0.8 on a raw scale and an alpha of 0.05) treated mice with a BMS score greater than 3. All data are given as mean ± SEM. $n$ = biologically independent animals. The data can be found in S1 Data. BMS, Basso Mouse Scale; CSP, carbon nanosphere; SCI, spinal cord injury; TOST, two one-sided tests.

activation with stem cell grafts [25,27], self-assembling peptide-based biomaterials [28], or both, since these might provide the necessary tissue bridge and relays to allow for increased functional synapses underpinning neurological recovery. Additionally, while in our view, tissue bridges will need to be part of the equation for spinal repair, the synergism with additional neuronal extrinsic interventions aimed to further enhance plasticity such as CSPG inhibition could also be considered.

More broadly, extending upon previous findings showing that PTEN deletion [29] or antagonism [30] benefit axonal growth and spinal circuitry formation in chronic SCI, the present work suggests that it is possible to promote axonal growth and plasticity in a chronic spinal cord injury leading to severe disability. Finally, similarly to TTK21, it encourages to attempt other molecular regenerative interventions, ideally in combination with tissue bridging approaches.

## Materials and methods

### Mice

Mouse experimentation was carried out in accordance with regulations of the UK Home Office under the Animals (Scientific Procedures) Act 1986, with Local Ethical Review by the Imperial College London Animal Welfare and Ethical Review Body Standing Committee (AWERB, PPL P6EDD65B1). Female C57Bl6 (Harlan, UK) mice ranging from 6 to 8 weeks of age were used for all experiments. For all surgeries, mice were anesthetized with isoflurane (4% induction 2% maintenance) and buprenorphine (0.1 mg/kg) and carprofen (5 mg/kg) were administered peri-operatively as analgesic.

Animals were kept on a 12-hour light/dark cycle with food and water provided ad libitum, at a constant room temperature and humidity (21˚C and 5%, respectively). SH for mice consists of $26 \times 12 \times 18$ cm$^3$ cages housing 4 mice with tissue paper for bedding, a tunnel, and a wooden chew stick. The EE housing consists of $36 \times 18 \times 25$ cm$^3$ cages housing 5 mice with tissue paper for bedding, a tunnel, and a wooden chew stick. EE cages also host the following: additional nesting material that included nestlets, rodent roll, and sizzle pet (LBS Biotech); a hanging plastic tunnel (LBS Biotech) and a plastic igloo combined with a fast-track running wheel (LBS Biotech); a wooden object (cube, labyrinth, tunnel, corner 15) (LBS Biotech) that is changed every 5 days to help maintain a novel environment; and 15 g fruity gems (LBS Biotech) every 5 days to encourage exploratory and natural foraging behaviour.

### Spinal cord injury (SCI)

Surgeries were performed as we previously reported [24]. A laminectomy at vertebra T9 was performed to expose spinal level T9 and a deep dorsal transection past the central canal at a depth of approximately 1.1 mm was carried out using microscissors (Fine Science Tools). This lesion only spares a portion of the ventral white matter and leaves the animals with a severe and permanent neurological impairment—no improvement from baseline. At week 22 after SCI, animals were deeply anesthetized and perfused transcardially.

### CSP-TTK21 administration

Animals were randomised to treatment after been subdivided into 2 groups with comparable severity based upon BMS with the CBP/p300 activator bound to carbon nanospheres (CSP-TTK21) or a control of just CSP (mice– 20 mg/kg injected IP once a week). Mice received the first IP injection 12 weeks after SCI until killing.

### Neuronal tracing

For dorsal column tracing of sensory DRG axons, 2 μl of Dextran-88 (Life Technology) was injected into the sciatic nerve bilaterally 1 week before killing using the fire polished glass capillary connected to a 10-μl Hamilton syringe.

For anterograde tracing of motor CSTs, 2 weeks before killing, mice that received an SCI were injected with the axonal tracer Dextran-tetramethylrhodamine and biotin, with a Hamilton microsyringe in the motor cortex following standard stereotaxic coordinates as previously described [31].

### 5-HT immunostaining

For 5-HT immunostaining, we followed a protocol we previously described [32]. The sections were incubated with rabbit anti-5-HT (1:500, Sigma S5545) in 4% NGS in 0.3% TBS-Triton

X100 for 4 days at 4˚C. Next, the sections were incubated with an Alexa fluorescent secondary antibody. Finally, the sections were coverslipped.

## Behavioral analysis

Mice were trained daily for 2 weeks preinjury before baseline measurements and then assessed on day 7 postinjury and biweekly thereafter; until CSP or CSP-TTK21 injections, mice were then tested weekly thereafter. All behavioral testing and analysis were done by 2 observers blinded to the experimental groups.

**Gridwalk.** Mice will cross a 1-m long horizontal grid 3 times. Videos of the runs were blindly analyzed at a later time point, and errors from both hind limbs were counted and normalized to total number of steps. Error values represent the total number of slips made by both hindlimbs over the 3 runs.

**Open-field test.** The BMS [33] was used to assess open-field locomotion. Each animal was allowed to freely move in the open field for 4 minutes while 2 independent investigators blinded to experimental group will score it. The BMS score and subscore were given. Only the animals showing frequent or consistent plantar stepping in the open field (BMS score $\geq$3) were tested on the grid walk.

**Von Frey.** The Von Frey test determines the mechanical force required to elicit a paw withdrawal response. Each animal was tested in each paw 3 times. Only animals showing plantar placement in the open field (BMS score $\geq$3) were tested.

**Hargreaves.** The Hargreaves test determines the latency of a thermal nociceptive stimulus required to elicit a paw withdrawal response. Each animal was in each paw 3 times. Only animals showing plantar replacement in the open field (BMS score $\geq$3) were tested.

## Quantification of axonal regeneration

For each spinal cord after injury, the number of fibres rostral and caudal to the lesion and their distance from the lesion epicentre (depending on whether sensory or motor axons) were analysed in 4 to 6 sections per animal with a fluorescence Axioplan 2 (Zeiss) microscope and with the software Stereo-Investigator 7 (MBF Bioscience). The lesion epicentre was identified by GFAP staining in each section at 20× magnification. The total number of labelled axons or signal intensity of the traced axons rostral to the lesion site were normalized to the total number of labelled axons or of the signal intensity caudal or rostral to the lesion site counted in all the analysed sections for each animal, obtaining an inter-animal comparable ratio. Sprouts and regrowing fibres were defined following the anatomical criteria reported by Steward and colleagues [34].

## Histology and immunohistochemistry

Tissue was postfixed in 4% paraformaldehyde (PFA) (Sigma) and transferred to 30% sucrose (Sigma) for 5 days for cryoprotection, the tissue was then be embedded in OCT compound (Tissue-Tek) and frozen at −80˚C. DRG, spinal cord, and cortices were sectioned at 10, 20, and 20 μm thickness, respectively, using a cryostat (Leica). Immunohistochemistry on tissue sections was performed according to standard procedures. For selected antibodies, antigen retrieval was performed submerging the tissue sections in 10 mM citrate buffer (pH 6.2) or 10 mM Tris/1 mM EDTA buffer (pH 9.0) at 98˚C for 5 minutes. Next, tissue sections were washed with PBS to remove the excess of citrate buffer and blocked for 1 hour with 8% BSA, 1% PBS-TX100. Finally, the sections were incubated with anti-p-STAT3 (1:100, Rabbit, Cell Signaling Technology 9145), c-Jun (1:100, CST, #9165), ATF3 (1:100, Santa Cruz, sc-188), pErk (1:250, CST, #9101), GAP43 (1:500, Sigma AB5220), KLF7 (1:100, Santa Cruz sc-

398576), SPRR1A (1:100, Thermo Fisher Scientific PA5-110423), Tuj1 (1:1,000, Promega G7121), H3K27ac (1:500, ab4729), H3K9ac (1:500, Cell Signalling 9671), GFAP (1:500, Millipore AB5804), CD68 (1:500, Abcam ab213363), vGlut1 (1:1,000, Synaptic system 135302), vGat1 (1:500, Synaptic systems 131011), ChAT (1:500, Sigma AB144P) antibodies at 4˚C O/N. This was followed by incubation with Alexa Fluor–conjugated secondary antibodies according to standard protocol (1:1,000, Invitrogen). Slides were counterstained with DAPI to visualise nuclei whenever necessary (1:5,000, Molecular Probes).

## Image analysis for IHC

All analysis was performed by the same experimenter who was blinded to the experimental groups. Photomicrographs were taken with a Nikon Eclipse TE2000 microscope with an optiMOS scMOS camera using 10× or 20× magnification using ImageJ (Fiji 64 bits 1.52 p), Micro-Manager 2.0 software for image acquisition or at 20× magnification with an Axioplan 2 (Zeiss) microscope and processed with the software AxioVision (Zeiss).

## Analysis of GFAP and CD68 intensity around the lesion site

GFAP and CD68 intensity and area with positive signal were quantified from sagittal spinal cord sections from 1 series of tissue for each animal. Quantification was done using ImageJ, the background was subtracted, and then the mean pixel intensity and area of immunoreactivity was measured.

## Analysis of fluorescence intensity

For quantitative analysis of pixel intensity, the nucleus or soma of DRG or layer V cortical neurons were manually outlined in images from 1 series of stained tissue for each mouse. To minimize variability between images, pixel intensity was normalized to an unstained area and the exposure time and microscope setting were fixed throughout the acquisition.

## Analysis of 5HT fibres in the ventral horn

Intensity of 5HT immunohistochemistry was measured in the ventral horn of L1-4 spinal sections. Quantification was done using ImageJ, the background was subtracted, and then the mean pixel intensity was measured from 1 series of tissue for each animal.

## Analysis of vGlut1 and vGat immunohistochemistry in proximity to motor neurons

vGlut1 and vGat synaptic boutons were imaged with a SP8 Leica confocal microscope. Z-stacks images were taken with an average thickness of 15 μm with a step size of 0.3 μm. Sequential line scanning was performed when more than 2 channels were acquired. Multifluorescent orthogonal 3D image analysis and visualization were performed using Leica LAS X software. The average number of vGlut1 or vGat boutons opposed to motor neurons in the ventral horn of L1-3 spinal sections was calculated by analysing 20 motor neurons per replicate. All analyses were performed blind to the experimental group.

## Statistical analysis

**Measures for avoidance of bias (e.g., blinding, randomisation).** We adhere to the principles of NC3Rs and adhere to the ARRIVE guidelines [35]. All experiments were performed in blind to the treatment and experimental group. Behavioral studies were assessed by 2 independent investigators (e.g., research associate and research assistant) blind to one another, to

the treatment, and to injury group when relevant. Randomization followed a computerized sequence. As far statistical analysis, our priority was to adopt a frequentist equivalence test. There are 5 parameters in our experimental design: sample size (n), difference between the groups (delta), standard deviation (sigma), type I error (alpha/significance), and type II error (power). We can use estimates for 4 of these parameters to calculate the 5th parameter—which we have done to achieve our required sample size to achieve a certain power and significance.

**Sample size calculation, power calculations, and justification of effect size.** The size of our in vivo experimental groups as planned in our aims has been defined following the Animal Experimentation Sample Size Calculator (AESSC) tool. Sample size calculations were performed using a two-tailed unpaired Student $t$ test and two-way repeated measures ANOVA (significance $\leq 0.05$; power $\geq 0.90$; G*Power); $N$ indicates number/group. The specific effect size has been estimated based upon similar studies showing significant differences between experimental and control group [8,21].

Two animals were added to each experiment based upon the probability of losing animals due to the experimental procedure such as spinal surgery. While the selected $N$ is derived by our power calculation, it is also compatible and comparable with what we have previously published for similar experiments [21]. Exclusion criteria include the following:

- Tissues with low quality for further experimentation or imaging as shown by lack of clear injury site or by lack of clear axonal tracing. Degraded cords in case of inefficient fixation or damaged during cryostat sectioning.

- Animals with injury severity that differs 2 SD from the mean as defined by the size of the injury site, which has to be within 2 SDs from the average.

No animal replacement was needed to ensure that the power requirements were met. Specific calculations are found here below:

Immunofluorescence (control vs. TTK21): change in fluorescence intensity of 1.5-fold (between 50 and 75 arbitrary units for example); SD = 10 and 12; P: 0.05; Power 90%; $N = 4$

Axonal regeneration (control vs. TTK21): increase in % of axons regenerating beyond the lesion site from 5 to 10; SD = 2 and 4; Power 90%; P: 0.05, N: 8

Behavioral tests (gridwalk as example test, control vs. TTK21): changes in score from 4 to 2; SD: 2 and 1; Power 90%; P: 0.05; $N = 12$

Results were expressed as mean ± SEM. Statistical analysis was carried out using GraphPad Prism 9. Normality was tested for using the Shapiro–Wilk test. Normally distributed data were evaluated using a two-tailed unpaired Student $t$ test or a two-way repeated measures ANOVA when experiments contained more than 2 groups. The Tukey or Sidak post hoc test was applied when appropriate. The Mann–Whitney $U$ test was used for nonparametric evaluation. Given the level of nonsignificance in behavioural experiments, a frequentist equivalence test using the two one-sided tests (TOST) rule was needed to strengthen the null hypothesis conclusion (TOSTER 0.4.1, parameters 0.18.1.6, R 4.1.2).

A threshold level of significance α was set at $P$ value $<0.05$. Significance levels were defined as follows: * $P$ value $< 0.05$; ** $P$ value $< 0.01$; *** $P$ value $< 0.001$, **** $P$ value $< 0.0001$. All data analyses were performed blind to the experimental group.

**URL to this deposited stage 1 manuscript**: 10.17605/OSF.IO/S5EDH

## Supporting information

**S1 Fig. (A)** Twelve weeks old mice were housed in SH or EE 1 week after a spinal cord transection. (**B** and **C**) Animals in SH remained impaired unable to step until day 42 after injury as shown by BMS (**B**) and Gridwalk (**C**). EE significantly enhanced locomotion (mean ± SEM,

two-way ANOVA, Fisher LSD post hoc [**] $P < 0.01$; [***] $P < 0.005$; [****] $P < 0.001$). The data can be found in S1 Data. BMS, Basso Mouse Scale; EE, enriched environment; SH, standard housing.
(PDF)

**S2 Fig. Graphical diagram summarizing the experimental design.** *Made with BioRender.*
(PNG)

**S3 Fig. (A)** Representative micrographs of cJUN immunostaining (green, white arrows) in layer 5 cortical neurons from CSP or CSP-TTK21-treated mice. (**B**) Quantification of cJUN immunostaining in layer 5 cortical neurons (CSP: 2,539.0 ± 295.4; CSP-TTK21: 2,355.0 ± 206.7, $p = 0.63$, TOST: t(0.1) = 0.1, $p = 0.5$ given equivalence bounds of −153.0 and 153.0 on a raw scale and an alpha of 0.05, $n = 4$). (**C**) Representative micrographs of SPRR1a immunostaining (green, white arrows) in layer 5 cortical neurons. (**D**) Quantification of SPRR1a immunostaining in layer 5 cortical neurons in CSP or CSP-TTK21 (CSP: 3,940.0 ± 491.6, $n = 3$; CSP-TTK21: 3,636.0 ± 350.3, $n = 4$; $p = 0.08$, TOST: t(4.8) = −1.9, $p = 0.94$ given equivalence bounds of −344.4 and 344.4 on a raw scale and an alpha of 0.05). Mean ± SEM; unpaired two-tailed Student *t* test or Welch *t* test. $n$ = biologically independent animals. The data can be found in S1 Data. CSP, carbon nanosphere; TOST, two one-sided tests.
(PNG)

**S4 Fig. (A)** Representative micrographs of GFAP intensity (red) around the SCI site (white asterisks) and cavity size (white dotted line) in CSP or CSP-TTK21 mice. (**B**) Quantification of cavity size in CSP or CSP-TTK21-treated mice (CSP: 471,574.0 ± 76,631.0, $n = 8$; CSP-TTK21: 486,466.0 ± 45,491.0, $n = 14$; $p = 0.87$; TOST: t(20.0) = 0.5, $p = 0.31$ given equivalence bounds of −56,344.2 and 56,344.2 on a raw scale and an alpha of 0.05). (**C**) Quantification of GFAP intensity in CSP or CSP-TTK21-treated mice *(CSP: 1,272.0 ± 31.3, $n = 3$; CSP-TTK21: 1,339.0 ± 20.8, $n = 4$, $p = 0.12$, TOST: t(3.7) = −1.4, $p = 0.88$ given equivalence bounds of −14.5 and 14.5 on a raw scale and an alpha of 0.05)*. Mean ± SEM; unpaired two-tailed Student *t* test or Welch *t* test. $n$ = biologically independent animals. The data can be found in S1 Data. CSP, carbon nanosphere; SCI, spinal cord injury; TOST, two one-sided tests.
(PNG)

**S5 Fig. (A)** Representative micrographs of CD68 immunofluorescence (red) and DAPI (blue) around the SCI site (white asterisks) in CSP or CSP-TTK21-treated mice. Lesion site (white dotted line). (**B**) Quantification of CD68 intensity in CSP or CSP-TTK21-treated mice (CSP: 472.0 ± 8.1, $n = 3$ CSP-TTK21: 434.5 ± 37.8 $n = 4$, $p = 0.44$, TOST: t(3.3) = 0.5, $p = 0.69$ given equivalence bounds of −16.3 and 16.3 on a raw scale and an alpha of 0.05). Mean ± SEM; unpaired two-tailed Student *t* test. $n$ = biologically independent animals. The data can be found in S1 Data. CSP, carbon nanosphere; SCI, spinal cord injury; TOST, two one-sided tests.
(PNG)

**S1 Data. Excel spreadsheets containing the quantitative data for each experiment as described in the results and figure legends.**
(XLSX)

## Author Contributions

**Conceptualization:** Franziska Müller, Simone Di Giovanni.

**Formal analysis:** Franziska Müller, Francesco De Virgiliis, Guiping Kong, Luming Zhou, Alexandros Sanchez-Vassopoulos.

**Investigation:** Franziska Müller, Francesco De Virgiliis, Guiping Kong, Luming Zhou.

**Methodology:** Franziska Müller, Francesco De Virgiliis, Guiping Kong, Luming Zhou, Elisabeth Serger, Jessica Chadwick, Akash Kumar Singh, Tapas K. Kundu.

**Project administration:** Simone Di Giovanni.

**Resources:** Akash Kumar Singh, Muthusamy Eswaramoorthy, Tapas K. Kundu, Simone Di Giovanni.

**Supervision:** Elisabeth Serger, Jessica Chadwick, Simone Di Giovanni.

**Writing – original draft:** Simone Di Giovanni.

**Writing – review & editing:** Franziska Müller, Francesco De Virgiliis, Luming Zhou, Tapas K. Kundu.

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
