## [Editor Report · Decision Letter 0]

1 Jun 2021

Dear Simone, 

Thank you for submitting your manuscript entitled "Combinatorial small molecule-mediated activation of CBP/p300 with environmental enrichment in chronic severe experimental spinal cord injury to enable axon regeneration and sprouting for functional recovery" for consideration as a Preregistered Research Article by PLOS Biology. Please accept my apologies for the delay in sending the decision below to you.

Your manuscript has now been evaluated by the PLOS Biology editorial staff. We have also discussed your proposal with two academic editors, one with expertise in the biological questions you are addressing and another one with expertise in Pre-registered Reports. I am writing to let you know that we are interested in peer-reviewing your proposal, but before we can do that, we would like you to address some concerns raised by the Academic Editor with expertise in Pre-registered Reports. These issues are very likely to come up with the reviewers and so, we think, addressing them now will save you time in the end. You can find the comments from the Academic Editors below my signature. When you re-submit, please provide a point-by-point response to her/his concerns. The Academic Editor is willing to answer any question you might have during the revision process. I would also be happy to answer any questions, via e-mail or phone/zoom.

In addition, and BEFORE you start your revision, we need you to complete your submission by providing the metadata that is required for full assessment. To this end, please login to Editorial Manager where you will find the paper in the 'Submissions Needing Revisions' folder on your homepage. Please click 'Revise Submission' from the Action Links and complete all additional questions in the submission questionnaire. (YOU DONT NEED TO SUBMIT YOUR REVISION YET, JUST COMPLETE THE METADATA). 

Please re-submit your manuscript with your metadata within two working days, i.e. by Jun 03 2021 11:59PM.

Once your full submission is complete, your paper will undergo a series of checks. Once they are complete, I will stamp a Major Revision decision to give you time to address the concerns below.

Feel free to email us at plosbiology@plos.org or ggasque@plos.org if you have any queries relating to your submission.

Kind regards,

Gabriel Gasque

Senior Editor

PLOS Biology

ggasque@plos.org

Academic Editor’s comments:

What is the source of the results presented in Supporting Figure 1? Is this pilot data or from a previous published article?

Please make clearer which outcomes would disconfirm the hypothesis, and for more complex hypotheses in which there are multiple measured variables, what strength of confirmation or disconfirmation would be associated with what combination of outcomes?

Please provide all G*Power outputs (e.g. screenshots of the output screen) as I couldn't reproduce all of these calculations.

On p17 the authors note that the “specific effect size has been estimated based upon similar studies showing significant differences between experimental and control group.” More detail is required here for a Stage 1 Registered Report. The studies and effect size estimates that furnish these estimates need to be sourced, e.g. in a table, listing the test and effect size reported in each previous study. Given that the target effect sizes are based on existing research, the authors should also be sure to take into account the effects of selection bias and publication bias, which typically inflate published effect size estimates? As note in the RR guidelines, since publication bias over-inflates published estimates of effect size, power analysis must be based on the *lowest* available or meaningful estimate of the effect size (i.e. the lower end of the effect size distribution).

For the hypotheses involving gridwalk and BMS data, the authors propose a repeated measures ANOVA, but it isn’t clear what factors are included in this analysis. The framing of the hypothesis in the design table instead suggests it would be tested through a pairwise group comparison. If there is a factor of group as well as a repeated measures factor, then presumably the authors instead intend to use a mixed ANOVA, not a repeated measures ANOVA.

To maximise clarity of the study procedure, I recommend including a schematic/figure that depicts the sequence of interventions and measurements for the two groups.

As I understand it from pp7-8, the control group will also be exposed to an enriched environment (EE) but will receive a control intervention (nanoparticles and vehicle alone). Given that the key comparison is (CSP-TTK21 + EE) vs (control treatment + EE), and EE is therefore held constant between the groups, how is it possible for the design to reveal any *combined* effects of CSP-TTK21 and EE, as opposed to testing the effects of CSP-TTK21 alone?

Update the design table to make clear the sample size that will be recruited based on the alpha level that will be used to conclude support for the hypothesis (e.g. do not report two sample sizes, for p<.05 for p<.01, as it is unclear which sample size will be included).

The authors note in the introduction that “lack of validation of this hypothesis will also provide essential information allowing the scientific community to entertain alternative hypotheses.” Since statistically non-significant results can only provide weak evidence for invariance between conditions (i.e. absence of evidence rather than evidence of absence), if the authors want to be able to draw stronger conclusions in the event of non-significant results, they may wish to consider frequentist equivalance tests (https://journals.sagepub.com/doi/10.1177/2515245918770963) or the use Bayes factors (https://www.frontiersin.org/articles/10.3389/fpsyg.2014.00781/full). Unlike conventional NHST, these tests can provide positive evidence of no effect.

One of the key criteria that reviewers are asked to assess in Stage 1 RRs is "Whether the authors have pre-specified sufficient outcome-neutral tests for ensuring that the results obtained are able to test the stated hypotheses, including positive controls and quality checks.", and successfully passing such tests is an editorial criterion at Stage 2 following completion of the study. Your protocol does not obviously propose any such tests - therefore please consider whether such positive controls or data quality checks are appropriate, and if possible how they might be included in the design. For instance, what positive control might be included (independently of the main hypotheses) to confirm that the CSP-TTK21 treatment was administered successfully? To consider what control is the most appropriate: imagine you ran the study and found null or confusing results. What positive control would convince a skeptic that the intervention was administered with sufficient precision and reliability to be able to provide a fair test of the hypothesis?

p12 mentions sham surgery, but it is unclear how this factors into the study design. My understanding from introduction was that *all* mice will receive the spinal cord injury?

On p17 the authors note that “two animals will be added to each experiment based upon the probability of losing animals due to the experimental procedure such as spinal surgery.” What happens if more animals are lost than anticipated? Will they be replaced until the minimum sample size is achieved, or there is a hard limit on the number of animals that can be tested?

Please ensure that exclusion criteria for data are comprehensively and precisely pre-specified as it usually not possible to adjust these for preregistered analyses after provisional acceptance is granted. At present the only reference to exclusion criteria is pp17-18: “Exclusion criteria include tissues with low quality for further experimentation or imaging as well as animals with injury severity beyond 2 SD from the mean.” This requires elaboration. How is “low quality” defined? Objectively or subjectively and at what point in the study timeline will this be assessed? How is injury severity defined? Why 2 SDs? Are excluded animals replaced to ensure the power requirements are met? Are there any other exclusion criteria of any kind? I suggest listing the exclusion criteria precisely in bullet point form and ensuring they are as exhaustive as possible.

---

## [Editor Report · Decision Letter 1]

4 Jun 2021

Dear Simone,

Thank you very much for submitting your manuscript "Combinatorial small molecule-mediated activation of CBP/p300 with environmental enrichment in chronic severe experimental spinal cord injury to enable axon regeneration and sprouting for functional recovery" for consideration as a Preregistered Research Article at PLOS Biology. As mentioned previously, manuscript has been evaluated by the PLOS Biology editors and by two Academic Editors with relevant expertise. One to the Academic Editors has provided some detailed feedback that we think you should address before we can send your paper to independent reviewers.

We anticipate addressing these issues should not take you very long. Thus, we expect to receive your revised manuscript within 1 month. You can find the detailed comments below my signature.

**IMPORTANT - SUBMITTING YOUR REVISION**

Your revisions should address the specific points made by the Academic Editor. Please submit the following files along with your revised manuscript:

1. A 'Response to Reviewers' file - this should detail your responses to the editorial requests, present a point-by-point response to all of the comments, and indicate the changes made to the manuscript. 

*Resubmission Checklist*

*Published Peer Review*

*PLOS Data Policy*

*Blot and Gel Data Policy*

Sincerely,

Gabriel Gasque

Senior Editor

PLOS Biology

ggasque@plos.org

Academic Editor's comments:

What is the source of the results presented in Supporting Figure 1? Is this pilot data or from a previous published article?

Please make clearer which outcomes would disconfirm the hypothesis, and for more complex hypotheses in which there are multiple measured variables, what strength of confirmation or disconfirmation would be associated with what combination of outcomes?

Please provide all G*Power outputs (e.g. screenshots of the output screen) as I couldn't reproduce all of these calculations.

On p17 the authors note that the “specific effect size has been estimated based upon similar studies showing significant differences between experimental and control group.” More detail is required here for a Stage 1 Registered Report. The studies and effect size estimates that furnish these estimates need to be sourced, e.g. in a table, listing the test and effect size reported in each previous study. Given that the target effect sizes are based on existing research, the authors should also be sure to take into account the effects of selection bias and publication bias, which typically inflate published effect size estimates? As note in the RR guidelines, since publication bias over-inflates published estimates of effect size, power analysis must be based on the *lowest* available or meaningful estimate of the effect size (i.e. the lower end of the effect size distribution).

For the hypotheses involving gridwalk and BMS data, the authors propose a repeated measures ANOVA, but it isn’t clear what factors are included in this analysis. The framing of the hypothesis in the design table instead suggests it would be tested through a pairwise group comparison. If there is a factor of group as well as a repeated measures factor, then presumably the authors instead intend to use a mixed ANOVA, not a repeated measures ANOVA.

To maximise clarity of the study procedure, I recommend including a schematic/figure that depicts the sequence of interventions and measurements for the two groups.

As I understand it from pp7-8, the control group will also be exposed to an enriched environment (EE) but will receive a control intervention (nanoparticles and vehicle alone). Given that the key comparison is (CSP-TTK21 + EE) vs (control treatment + EE), and EE is therefore held constant between the groups, how is it possible for the design to reveal any *combined* effects of CSP-TTK21 and EE, as opposed to testing the effects of CSP-TTK21 alone?

Update the design table to make clear the sample size that will be recruited based on the alpha level that will be used to conclude support for the hypothesis (e.g. do not report two sample sizes, for p<.05 for p<.01, as it is unclear which sample size will be included).

The authors note in the introduction that “lack of validation of this hypothesis will also provide essential information allowing the scientific community to entertain alternative hypotheses.” Since statistically non-significant results can only provide weak evidence for invariance between conditions (i.e. absence of evidence rather than evidence of absence), if the authors want to be able to draw stronger conclusions in the event of non-significant results, they may wish to consider frequentist equivalance tests (https://journals.sagepub.com/doi/10.1177/2515245918770963) or the use Bayes factors (https://www.frontiersin.org/articles/10.3389/fpsyg.2014.00781/full). Unlike conventional NHST, these tests can provide positive evidence of no effect.

One of the key criteria that reviewers are asked to assess in Stage 1 RRs is "Whether the authors have pre-specified sufficient outcome-neutral tests for ensuring that the results obtained are able to test the stated hypotheses, including positive controls and quality checks.", and successfully passing such tests is an editorial criterion at Stage 2 following completion of the study. Your protocol does not obviously propose any such tests - therefore please consider whether such positive controls or data quality checks are appropriate, and if possible how they might be included in the design. For instance, what positive control might be included (independently of the main hypotheses) to confirm that the CSP-TTK21 treatment was administered successfully? To consider what control is the most appropriate: imagine you ran the study and found null or confusing results. What positive control would convince a skeptic that the intervention was administered with sufficient precision and reliability to be able to provide a fair test of the hypothesis?

p12 mentions sham surgery, but it is unclear how this factors into the study design. My understanding from introduction was that *all* mice will receive the spinal cord injury?

On p17 the authors note that “two animals will be added to each experiment based upon the probability of losing animals due to the experimental procedure such as spinal surgery.” What happens if more animals are lost than anticipated? Will they be replaced until the minimum sample size is achieved, or there is a hard limit on the number of animals that can be tested?

Please ensure that exclusion criteria for data are comprehensively and precisely pre-specified as it usually not possible to adjust these for preregistered analyses after provisional acceptance is granted. At present the only reference to exclusion criteria is pp17-18: “Exclusion criteria include tissues with low quality for further experimentation or imaging as well as animals with injury severity beyond 2 SD from the mean.” This requires elaboration. How is “low quality” defined? Objectively or subjectively and at what point in the study timeline will this be assessed? How is injury severity defined? Why 2 SDs? Are excluded animals replaced to ensure the power requirements are met? Are there any other exclusion criteria of any kind? I suggest listing the exclusion criteria precisely in bullet point form and ensuring they are as exhaustive as possible.

---

## [Decision Letter · Decision Letter 2]

28 Jul 2021

Dear Simone,

Thank you for submitting your manuscript "Small molecule-mediated activation of CBP/p300 with environmental enrichment to enable axon regeneration and sprouting for functional recovery in chronic severe experimental spinal cord injury" for consideration as a Preregistered Research Article at PLOS Biology. Your manuscript has been evaluated by the PLOS Biology editors, by an Academic Editor with relevant expertise, and by three independent reviewers. You will note that reviewer 3 has revealed his identity. 

In light of the reviews (below), we will not be able to accept the current version of the manuscript, but we would welcome re-submission of a much-revised version that takes into account the reviewers' comments. We cannot make any decision about publication until we have seen the revised manuscript and your response to the reviewers' comments. Your revised manuscript is also likely to be sent for further evaluation by the reviewers.

We expect to receive your revised manuscript within 3 months. 

**IMPORTANT - SUBMITTING YOUR REVISION**

Your revisions should address the specific points made by each reviewer. Having discussed these comments with the academic editors, we would like you to consider the following:

1) Regarding reviewer 3's recommendation to use rats instead of mice, we do not think that it is necessary that you switch your model system as recommended. However, you should be reserved in your discussion of the true clinical impact of your work.

2) We agree with reviewer 3 that four groups should be tested for this study, even though previous studies have completed two of the groups. 

3) We also think that negative results may be difficult to interpret. Thus, we would recommend that you are reserved in the conclusions you draw from the study given that hierarchy of synergy is not thoroughly tested. You may be able to address this with a clearer rationale of the treatment paradigm, as was requested by all the reviewers, but see point (5) below. 

4) We agree with the concerns of the reviewers regarding positive controls and sample size, which go to the heart of the criteria for accepting a Registered Research Articles Stage 1. It will be important for you to thoroughly address these points (among the many others).

5) We would also like to see a more comprehensive specification of the proposed frequentist equivalence tests and/or Bayesian hypothesis tests (including, especially, the choice of prior for the Bayesian tests and justification of that prior) and the chosen parameters for the frequentist tests. You also need to be clear which tests, Bayesian or frequentist equivalence, will determine the interpretation in the event of negative results, as it is not guaranteed that they will produce equivalent strength of evidence.

Please submit the following files along with your revised manuscript:

1. A 'Response to Reviewers' file - this should detail your responses to the editorial requests (above), present a point-by-point response to all of the reviewers' comments, and indicate the changes made to the manuscript. 

*Re-submission Checklist*

*Published Peer Review*

*PLOS Data Policy*

*Blot and Gel Data Policy*

Sincerely,

Gabriel Gasque

Senior Editor

PLOS Biology

ggasque@plos.org

REVIEWS:

Reviewer #1: This study proposes to investigate whether the epigenetic strategies to stimulate regenerative gene expression program combined with neuronal activity-dependent enhancement of neuroplasticity and guidance can overcome the current inability to promote neurological recovery in severe and chronic spinal cord injury. Specially, the authors propose to deliver the small molecule CBP/p300 activator CSP-TTK21 in mice housed in an enriched environment.

The plan is to administer the CBP/p300 activator CSP-TTK21 (i.p.) once a week between week twelve and twenty following a severe transection model of spinal cord injury in the mouse. A control group of mice will be treated with nanoparticles and vehicle alone. The CSP-TTK21 and control nanoparticles will be used at the dosage of 20mg/kg that showed efficacy in subacute spinal cord injury as published recently.

Data analysis will assess modifications in regenerative signalling, in motor and sensory axon sprouting and regeneration, in synaptic plasticity as well as in neurological sensorimotor recovery. 

Specifically, sprouting and regeneration of the dorsal columns will be analyzed with the retrograde axonal tracer Dextran (injected in the sciatic nerve in proximity of L4-L6 DRGs 7 days before sacrificing the animals). Sprouting and regeneration of the corticospinal

tracts (CSTs) will be analysed with stereotaxic injections of the neural tracer BDA into the

motor cortex 2 weeks before sacrifice. The authors will measure sprouting and regeneration of serotoninergic raphe-spinal motor tracts with 5-HT immunohistochemistry. To assess whether the CSP-TTK21 treatment will enhance synaptic plasticity, we will measure the number of inhibitory VGAT or excitatory VGLUT1/2 synaptic terminals in proximity of neuronal targets

such as interneurons in the dorsal horns and motoneurons in the ventral horns of the spinal cord (ChAT or NeuN immunostaining) including in association with specific tracing of CST,

sensory or 5-HT fibres. 

Histone acetylation as read out of CBP/p300 activation will also be evaluated in layer V neurons, raphe nuclei and DRG neurons by immunofluorescence. The expression of several regeneration associated genes including ATF3, JUN, GAP43, SPRR1a, KLF family members, and STAT3 will also be studied by immunofluorescence in sensory and motor neurons.

The authors will assess locomotion, coordination and sensorimotor integration by performing

open field assessment with the BMS and the gridwalk tests. In addition, Von Frey test for mechanoception and mechanical allodynia as well as Hargreaves test for thermoception and thermal hyperalgesia will be used to specifically assess the function of the ascending sensory tracts.

Previously, the authors have reported that the small molecule proposed here was able to promote sensory axon regeneration and recovery after a dorsal hemisection SCI in mice. In that study, injured mice received a weekly intraperitoneal injection of CSP-TTK21 (20 mg/kg) or control CSP, beginning 4 hours after injury. The current study seeks to address whether delayed treatment in chronically injured animals can also promote axon regeneration, sprouting and functional recovery. Considering the unmet need to ameliorate and restore functions in chronic SCI patients, this study addresses important aspect of SCI. The results obtained from the proposed experiment may provide insightful clues about developing potential reparative therapy, determine whether growth promoting strategies and enhancement of neural activity are an effective therapeutic strategy in chronic spinal cord injury.

The proposed experiments are technically sound and feasible. Methods and reagents to be used are described clearly with relevant citations. Statistical analyses to be performed seem appropriate. Exclusion criteria for animals is adequately described. The authors have significant experience and have published several manuscripts in the past with similar study design. 

There are only a few minor suggestions which could be addressed to further clarify the study.

-The authors had previously proposed that EE-induction of lasting increase in regeneration potential is mediated by a Cbp-dependent increase in histone acetylation and increase in gene expression, including pathways involved in neuronal activity, axonal projection and cytoskeleton remodeling. While EE likely affects multiple pathways and mechanisms that might act positively to improve SCI outcomes, the authors indicated that Cbp-dependent mechanisms likely mediate EE effects. As such, it is less clear why a strong synergy is expected when EE and the drug are combined. 

-The sex of mice that will be used in this study is not mentioned. 

-Expected timeline for the completion of the study was not included but would be helpful.

-The author might want to consider using Minimum Information about a Spinal Cord Injury Experiment (PMID: 24870067).

Reviewer #2: The proposed study takes an innovative and novel approach to a problem of very high importance. Coaxing repair from the central nervous system after injury is a long-standing challenge, and most work has focused on so-called acute interventions, which are applied immediately after injury. This fails to address the needs of millions of individuals with existing injuries, and even for future injuries the medical reality is that acute treatment is not always possible. The proposal here is to combine an enriched environment with a pharmacological treatment that targets an epigenetic mechanism, which will be administered many weeks after a spinal injury. A comprehensive battery of outcome measurements will then determine whether this treatment improves axon growth, animal behavior, and/or synapse formation in the damaged spinal cord.

The methodology and plans for analysis are relatively standard for the field and are well described, which raises confidence that the study will yield useful data (positive or negative) as planned.

There are some conceptual and technical issues that arise. I emphasize again the potential importance and novelty of this study; the comments below are offered in the spirit of helpful refinement or simply providing the authors a chance to help me better understand the rationale for some details.

1. The largest conceptual question: what is the rationale for supplying enriched environments one week after injury, but delayed CBP stimulation? And the closely related question: if the model is that EE is acting through a CBP mechanism, what is the rationale for providing EE to all animals and then additional CBP to half? The concern is that if EE already maximally engages the CBP-dependent pathway, there will be no added benefit to the pharmacological treatment. In general there is potential confusion here about the relationship between the mechanisms of the two treatments, how that relationship affects the rationale of their dual use, and how that relationship affects the selection of their timing.

2. Why is the CST being traced and analyzed, as opposed to the Gi as previously? The Gi is likely more relevant to locomotion, and there is already data to support some effect. The jump to CST is more likely to yield negative results and less likely to be related to the behaviors in question.

3. The study seems to lack positive controls. Given the extreme challenge of promoting axon growth in the chronic injury condition, negative results are quite possible. To plan for this, it seems important to have positive controls in place, especially for the histone and gene expression measurements.

4. A smaller technical question - why BDA for tracing CST axons? Viral tracing methods, used previously by this group, would seem to offer more sensitive detection of fine collaterals in the spinal grey matter. But perhaps this is not the case, or there may be other technical considerations that led to the selection of BDA? 

Reviewer #3, Mark H. Tuszynski: This is an interesting proposal to study a combination of a slow-release nanoparticle to activate CREB-binding protein (CSP-TTK21) and environmental enrichment (EE) in mice after spinal cord injury (SCI). The work is proposed by a careful group that publishes high quality, well-documented studies in high impact journals.

 The Di Giovanni group has previously published that EE enhances sensory axonal growth and motor behavioral recovery after SCI: EE increases H4K8ac protein levels and phosphorylation of CREB (Hutson TH et al, Sci Transl Med, 2019), as shown in Supporting Fig 1.

 I have one major comment regarding the experimental design, several moderate comments, and some minor comments.

 Major: Experimental groups - there will only be two experimental groups even though 2 therapies will be applied. I believe that for the emerging data from this study to be clearest and of the highest quality, there should be 4 groups: 1) Untreated control lesioned. 2) Treatment with EE. 3) Treatment with TTK21. 4) EE + TTK21. The authors may consider their previous report regarding the benefit of EE alone to be sufficiently compelling that they need not show the value of this alone, but I do not agree. This is a different study and there may be different effect sizes of EE, and synergies between EE and TTK21 that are only evident by examining adequate controls for each treatment.

 Moderate:

 Sampling Plan: The authors will study only 6 animals per group based on a power calculation that assumes a 200% increase in axon growth between the control and treatment group. This is very optimistic! I strongly suggest that the authors consider increasing the N per group to 12. 

 The functional studies also propose an effect size of 100%, which once again is quite optimistic (notwithstanding the preliminary data in Supp Fig 1). Again, I would increase group size to 12.

 The authors propose to use the mouse model. Why use this small animal species as opposed to the rat? Data in rats may be more clinically relevant, and the authors state that they wish to design a clinically relevant experiment.

 The authors reference "chronic" injury at several points of the paper, but EE is initiated one week after the injury in the proposed studies, not late, and a 3 month post-injury time point is not in any case necessarily "chronic". The authors might consider relying less heavily on this term. Perhaps more importantly, the authors might consider starting BOTH therapies 3 months after injury, or at least starting EE one month after injury: the average time in the U.S. that most patients enter rehabilitation is one month after injury, and few begin rehab one week after injury. But if the authors truly wish to study a late time point after injury when immediate post-injury events have subsided, initiating both therapies at 3 months would be better, in my opinion.

Minor:

 The authors state that they will use a "severe" transection model of SCI, but a dorsal over hemisection is not that severe. Mouse with this injury spontaneously recover to a BMS score of nearly 3, are show a gradual improvement in function over 42 days, although the authors state that this model causes "permanent" deficits. These lesions spare the ventral motor tracts of the spinal cord (esp reticulospinal) which can support several motor behaviors. These statements do not entirely correspond to the presented data in Supporting Figure 1, and some clarification is needed both with regard to exactly what the lesion is, and why it is considered permanent.

 The authors propose to use BDA as a tracer of the corticospinal system. Generally, our experience has been that newer fluorescent tracers such as TdTomato or membrane-trafficked tracers expressing fluorophores provide superior corticospinal axon visualization. The authors might consider using these tracers.

 The authors propose to perform immunolabeling for a number of RAGs to gauge expression levels: this is some variability in these immunolabeling methods. Might it be possible to pursue a less extensive list of RAGs and study levels by, e.g., Western blot of quantitative in situ hybridization?

 The authors inconsistently state that they will trace central projections of DRG axons with CTB in one part of the description, and Dextran red in another.

 The authors provide supporting data in support of EE enhancing motor recovery. However, they do not provide anatomical data that supports their claim of axonal/synaptic reorganization. According to the text, this should be included in supporting figure 1.

 For 5-HT immunolabelling, it is probably unnecessary to post-fix in glut, then perform antigen retrieval. In our experience, anti-5-HT antibodies (from Immunostar) perform well with 4% pfa fix without antigen retrieval.

 Behavioral analysis: Assessments will be performed on week 1 post-injury and weekly thereafter, and CSP-TTK21 injections will be performed on the same day. I suggest that the motor assessments be performed prior to the IP injections.

 Interpretations of negative outcomes: The authors very nicely propose several possible interpretations of negative outcomes (i.e., a lack of anatomical, behavioral or molecular differences from controls). But since this is a combinatorial treatment paradigm, there may in fact be many additional reasons that the approach could yield data. While all possibilities for a negative outcome are likely to be elucidated in this study, it remains worthy of study.

Overall, this is an interesting study that appears likely to yield useful and potentially important data. I hope that the preceding suggestions are helpful.

---

## [Decision Letter · Decision Letter 3]

14 Sep 2021

Dear Simone,

Thank you for submitting your revised Preregistered Research Article entitled "Small molecule-mediated activation of CBP/p300 with environmental enrichment to enable axon regeneration and sprouting for functional recovery in chronic severe experimental spinal cord injury" for publication in PLOS Biology. I have now obtained advice from the original reviewers and have discussed their comments with the Academic Editor. 

Based on the reviews, we are prepared to accept this proposal and to invite you to move ahead and do the experiments. However, before we do that, we would like to give you the opportunity to consider and respond, if you wish, to the lingering concerns expressed by reviewer 2. Addressing these concerns is not a requirement for eventual acceptance.

Please also include within your manuscript the ID number of the protocol approved by the Imperial College London Animal Welfare and Ethical Review Body Standing Committee (AWERB).

We expect to receive your revised manuscript within two weeks. 

*Published Peer Review History*

Sincerely,

Gabriel Gasque, Ph.D.,

Senior Editor,

ggasque@plos.org,

PLOS Biology

Reviewer remarks:

Reviewer #1: The revision has addressed sufficiently the concerns and questions raised by this reviewer. 

Reviewer #2: The authors have modified and significantly strengthened the manuscript to address the main concerns raised in the prior round of review. Most of the points were convincingly addressed. The rationale for providing EE to all animals as a surrogate for the normally "enriched" environment that is available to human patients - or put another way, to undo the artificial deprivation that likely distorts results in many lab animals - is clear. Clarifying the technical rationale for focusing here on CST growth, and adding the possibility of tracing reticulospinal axons if possible, is a strong response. In the same way, the plans for axon tracing were clarified and strengthened. 

A few lingering concerns exist but may be minor. One could quibble and point out that the rationale offered for combining EE and CBP stimulation somewhat missed the question. The authors have clarified that EE does not act solely through CBP; that is quite clear and was not the source of confusion. The question was the opposite - are we sure that CBP activation via CSP-TTK21 is offering a benefit beyond the stimulation of CBP already achieved through EE? In other words, is the pharmacological effect on CBP somehow stronger or broader than EE's effect on CBP? If not, the drug is simply duplicating an effect already present in EE animals. But on reflection, a stronger effect from the drug seems likely enough that the overall experiment is interesting and very worthwhile. I clarify this point only to sensitize the authors to the likelihood that other readers may wonder the same thing - not just whether EE is doing more than just activating CBP, which is now clarified in the revised manuscript, but also whether CSP-TTK21 is doing more to CBP than EE alone, which does not seem to be addressed. It would seem that both must be true to predict synergy.

 Finally, in response to questions about positive controls for histone and gene expression outcomes, the authors reiterated the intention to measure histone acetylation, pointing to prior findings that CSP-TTK21 treatment increased H4K8ac. Again, this was understood when the question was raised. This prior finding was obtained by acute delivery, not in the chronic state, so even in a narrow sense it is not clear that this readout qualifies as a positive control per se; it seems more of an open question, whether the histones of chronically injured neurons will be modified in the same way. More broadly, it leaves unanswered any question about positive controls for the RAGs to be measured. In reading responses to other reviewers, it appears that the intention is to compare the immunofluorescence obtained in these samples to other datasets obtained previously in the lab. In light of the outstanding and extensive track record of the lab, this is probably acceptable, although of course it must be done cautiously. 

Reviewer #3, M Tuszynski: The authors have responded well to my comments. No additional suggestions.

---

## [Editor Report · Decision Letter 4]

17 Sep 2021

Dear Simone,

Thank you for submitting your revised manuscript "Small molecule-mediated activation of CBP/p300 with environmental enrichment to enable axon regeneration and sprouting for functional recovery in chronic severe experimental spinal cord injury" for consideration as a Preregistered Research Article at PLOS Biology. Your Stage 1 manuscript has now been evaluated by the PLOS Biology editors, who have determined that your Stage 1 Protocol meets our criteria for importance of research question and technical soundness of the study proposal. We would therefore like to invite you to complete the study, as proposed, and submit the Stage 2 manuscript. Please carefully read all the following information.

*Explanation of Decision*

This is a Stage 1 'in-principle acceptance' decision, with a commitment to publish the final Stage 2 Preregistered Research Article (after revision, if needed), pending successful completion of the study according to these Stage 1 approved methods and analytic procedures, as well as an evidence-based interpretation of the results. Please see here for review criteria for Stage 2 manuscripts:

https://journals.plos.org/plosbiology/s/reviewer-guidelines#loc-reviewing-preregistered-research-articles

Editorial decisions will not subsequently be based on the perceived importance or novelty of the results obtained during the Stage 2 study. It is critical however that you adhere exactly to this approved Stage 1 study design when performing the study. Any deviation from these experimental procedures could lead to rejection of the manuscript at Stage 2. Please consult the editors immediately for advice if you need to alter this approved study plan.

**IMPORTANT**: Please follow the link below for important information regarding the Stage 2 manuscript template and review criteria. Please carefully read the guidelines on Stage 2 data collection BEFORE performing your study and completing your Stage 2 manuscript. 

AUTHOR GUIDELINES: https://plos.io/AuthorGuidelines

*Depositing this Stage 1 Protocol*

PLOS Biology does not publish Stage 1 Protocols immediately following an in-principle acceptance. Instead they are held and integrated into a single, completed 'Preregistered Research Article' following review and acceptance of the final Stage 2 manuscript. You are however required to register this approved Stage 1 Protocol with the Center for Open Science (https://cos.io/prereg/) or another recognised repository. This may be done publicly or under private embargo until submission of the Stage 2 manuscript. Stage 1 Protocols can be quickly and easily registered using a tailored mechanism for Registered Reports (https://osf.io/rr/). Please do this now. You will need to include the URL to this deposited protocol in your Stage 2 manuscript.

*Timeline*

We understand that carrying out the study will require a significant length of time and and are willing to allow you six months to perform the study. Please email us at 'plosbiology@plos.org' to discuss this if you have any questions or concerns, or to discuss an alternate timeline.

At this stage, your manuscript remains formally under active consideration at our journal. Please notify us by email if you do not wish to submit a Stage 2 manuscript or wish to pursue publication elsewhere, so that we may end consideration of the manuscript at PLOS Biology. 

*Resubmission Checklist*

Before submitting the Stage 2 manuscript, please review the following resubmission checklist: https://plos.io/Biology_Checklist

Please note that for PRA stage 2, the response to reviewers file does not follow the standard format, but should rather be a document for the reviewers detailing the changes made to the manuscript since the stage 1 accept.

*Published Peer Review*

*PLOS Data Policy*

Please note that as a condition of publication, PLOS' data policy (http://journals.plos.org/plosbiology/s/data-availability) requires that you make available all data used to draw the conclusions arrived at in your manuscript. Please note that for this article type, the raw data itself should be archived and made freely available in a public repository rather than submitted as supplementary material. Please make sure to read the Stage 2 submission guidelines online regarding how this data should be annotated and appropriately time stamped to show that data was collected after this Stage 1 in-principle acceptance and not before.

*Blot and Gel Data Policy*

To enhance the reproducibility of your results, we recommend that, if applicable, you deposit your laboratory protocols in protocols.io, where a protocol can be assigned its own identifier (DOI) such that it can be cited independently in the future. For instructions see: https://journals.plos.org/plosbiology/s/submission-guidelines#loc-materials-and-methods

Thank you again for your submission to PLOS Biology. We hope that our editorial process has been constructive thus far, and we welcome your feedback at any time. Please don't hesitate to contact us if you have any questions or comments.

Sincerely,

Gabriel Gasque

Senior Editor

PLOS Biology

ggasque@plos.org

---

## [Editor Report · Decision Letter 5]

15 Jun 2022

Dear Simone,

It was very nice meeting you last week at the Keystone meeting. I'm following up with you on your Stage 2 PRA submission at PLOS Biology. I've discussed the submission with both of the academic editors we are consulting on this work (one is on the study as a technical expert, one as our resident PRA guru). The technical academic editor is happy to have this go back out for peer review. In order to make the path as smooth as possible, the PRA academic editor has, however, asked if you could address the following points before we send it back out for Stage 2 peer review. They've said:

1. As per the RR policy authors are required to register the approved Stage 1 protocol on the OSF at the point of IPA and include the URL to the Stage 1 manuscript in the Stage 2 manuscript. Please add this.

2. Reading the results as a non-specialist, I'm not seeing as clear a mapping between the analyses and the hypotheses as I would have expected. I suggest the following to make this clear: (a) add a column to the design table called "Outcome and conclusion" which briefly states whether the hypothesis was supported or unsupported. (b) in the results, make clear which analyses are preregistered and which are not, and whether the hypothesis in each case is supported. I suggest structuring the results around the research questions in the design table so that the mapping is absolutely clear.

3. Deviations from protocol (or at least wording)

* First paragraph of Experimental Design section (p7). There appears to be a significant change in phrasing here. Does this reflect a change in wording or a change in the actual procedure?

* Is there any significance in the different colours of tracked changes? The tracked changes at this point should show all changes from the approved Stage 1 manuscript and *only* those changes. I haven't had time to check the approved Stage 1 manuscript against this tracked changes version to see if this is the case (a staff editor should do this prior to review and ensure everything is sound)

* Animals were sacrificed at 22 weeks post-injury rather than 20 weeks as preregistered. Why was this? A Deviations from Protocol seciton should be added to the Methods that lists, explains and justifies any and all changes from the approved procedures, however minor / inconsequential that are. Note: this only applies to changes in procedures, i.e. what was actually done, not minor text alterations.

4. I didn't understand the rationale for not doing the frequentist equivalence tests (p23). The authors suggest that the level of nonsignificance was of a sufficient level not to require it, but this is not statistically coherent. Without equivalence tests, statements such as "CBP/p300 activator TTK21 does not affect neurological recovery in chronic severe SCI" are statistically unfounded because they rely on non-significant results (i.e. they are only absence of evidence, not evidence of absence). All preregistered tests need to be reported.

====

**IMPORTANT - SUBMITTING YOUR REVISION**

*Resubmission Checklist*

*Published Peer Review*

*PLOS Data Policy*

*Blot and Gel Data Policy*

Sincerely,

Kris

Kris Dickson, Ph.D. (she/her)

Neurosciences Senior Editor/Section Manager

PLOS Biology

kdickson@plos.org

REVIEWS:

---

## [Decision Letter · Decision Letter 6]

5 Aug 2022

Dear Dr Di Giovanni,

Thank you for your patience while we considered your revised Stage 2 Preregistered Article "CBP/p300 activation promotes axon growth, sprouting and synaptic plasticity in chronic severe experimental spinal cord injury" for publication as a Preregistered Research Article at PLOS Biology. This revised version of your manuscript has been evaluated by the PLOS Biology editors, two Academic Editors (one topic-related; one to assess the Preregistration protocols) and the original reviewers.

Based on the reviews and our Academic Editor's assessment of your revision, we are likely to accept this manuscript for publication, provided you satisfactorily address the remaining points raised by the reviewers regarding the results and discussion. As this is a Preregistered Article (Registered Report), please disregard the comments from Reviewer 3 regarding suggested changes to the Introduction and Methods. Further, please ensure that the Introduction and Methods in the final revised manuscript match the prior approved Stage 1 Protocol manuscript. Textual changes should only be done to correct verb tense or, when necessary, to correct a factual error or to avoid a misunderstanding. 

Please also provide a blurb which, if the paper is accepted, will be included in our weekly and monthly Electronic Table of Contents (eTOCs), sent out to readers of PLOS Biology. This blurb may also be used to promote your article on social media. The blurb should be about 30-40 words long and is subject to editorial changes. It should, without exaggeration, entice people to read your manuscript, should not be redundant with the title and should not contain acronyms or abbreviations. For examples, view our author guidelines: https://journals.plos.org/plosbiology/s/revising-your-manuscript#loc-blurb

Finally, please also make sure to address the data and other policy-related requests listed at the bottom of this email. 

We expect to receive your revised manuscript within two weeks. 

*Published Peer Review History*

*Press*

Sincerely,

Kris

Kris Dickson, Ph.D. (she/her)

Neurosciences Senior Editor/Section Manager,

kdickson@plos.org,

PLOS Biology

PREREGISTERED ARTICLE POLICIES:

In addition to the comments listed above regarding the introduction and methods sections, please move the public Stage 1 Protocol URL to the Methods section: https://osf.io/s5edh

More details on our guidelines for Preregistered Articles can be found here: 

https://plos-marketing.s3.amazonaws.com/Marketing/Biology+Preregistered+Articles+Guidelines+for+Authors.pdf

DATA POLICY:

Note that we do not require all raw data. Rather, we ask that all individual quantitative observations that underlie the data summarized in the figures and results of your paper be made available. We appreciate the provision of this data in the supplementary file you've provided. 

We were not, however, able to locate the summary data for Supplemental Figure 1. 

***Please add this data to the excel document.

***Please also ensure that figure legends in your manuscript include information on where the underlying data can be found, and ensure your supplemental data file/s has a legend.

Please ensure that your Data Statement in the submission system also accurately describes where your data can be found.

DATA NOT SHOWN?

- Please note that per journal policy, we do not allow the mention of "data not shown", "personal communication", "manuscript in preparation" or other references to data that is not publicly available or contained within this manuscript. When going back over your study for final submission, please make sure to check for such statements, and either remove mention of any such data or provide figures presenting the results and the data underlying the figure(s).

Reviewer remarks:

Reviewer's Responses to Questions

Do you want your identity to be public for this peer review?

Reviewer #1: No

Reviewer #2: No

Reviewer #3: Yes: Mark Tuszynski

Reviewer #1: In this study, the authors investigated whether delayed delivery of CBP/p300 activator TTK21 in adult mice after severe transection SCI in combination with enhanced environment (EE) housing promotes histone acetylation, axonal and synaptic plasticity and behavioral recovery.

Sprouting and regeneration of the dorsal columns and CSTs were analyzed with the retrograde and anterograde axonal tracers, respectively. Axonal dieback as well as the

number of fibres past the lesion site were normalized to the number of labelled fibres prior to

the lesion. The authors measured sprouting of serotoninergic raphe-spinal motor tracts with 5-HT immunohistochemistry. To assess whether the CSP-TTK21 treatment enhances synaptic

plasticity, they measured the number of inhibitory vGat or excitatory vGlut1 synaptic terminals

in proximity of neuronal targets such as interneurons in the dorsal horns and motoneurons in

the ventral horns of the spinal cord (ChAT or NeuN immunostaining).

Histone acetylation as read out of CBP/p300 activation was evaluated in layer V neurons, raphe nuclei and DRG neurons by immunofluorescence. The expression of several regeneration associated factors including ATF3, JUN, GAP43, SPRR1a, KLF7, pERK, and pSTAT3 was studied by immunofluorescence in sensory and motor neurons.

Locomotion, coordination and sensorimotor integration were measured by performing open field assessment with the BMS and the gridwalk tests. Von Frey test for mechanoception and mechanical allodynia as well as Hargreaves test for thermoception and thermal hyperalgesia were used to specifically assess the function of the ascending sensory tracts.

The authors conclude that TKK21 treatment promotes histone acetylation in both DRG and cortical neurons, increases expression of some RAGs in DRGs, prevents axonal dieback, promotes axonal sprouting, particularly for 5-HT axons. However, recovery of function was not observed in these TTK21-treated mice. 

This study was set out to address an important question of whether previously identified regenerative treatment as shown in acute SCI models can promote plasticity and recovery in chronically injured animals. Importantly, the authors reason that EE does not represent a specific form of focused rehabilitation, but rather a more physiological setting compared to (standard housing) SH that better reflects the human condition where patients are encouraged to engage in physical activities after a spinal cord injury. There were several suggestions raised by the reviewers in the first round of review, most of which were decided not to be taken in the final study. These suggestions include inclusion of additional control groups, performing EE treatment months after SCI (i.e. at the time of TTK21 treatment) to reflect a delayed treatment paradigm, and choice of anterograde tracing method. However, it can be viewed that the authors present reasonable justification at least for the latter two recommendations. 

From the study, the authors conclude that although TTK21 with EE promotes histone acetylation and expression of select RAGs in sensory neurons, this treatment paradigm alone is insufficient to promote recovery of functions. The authors discuss that the chronically injured environment of CNS and spinal cord might need further modifications to successfully induce functional recovery. 

Th overall conclusion of the findings is an important addition to the SCI field and highlight the clear challenges for treating chronic SCI patients. The techniques used in this study are standard in the field and the authors have carried out carefully designed experiments to answer specific questions.

Minor points:

Figure 2A, there is a duplicate of NeuN image for the TTK21 group, and the ATF3 single channel image for the TTK21 group is missing. 

Figure 3A-D, it is unclear if the rostral axons represent those labeled axons that have regenerated past the lesion site (i.e. distal to the lesion). If so, it is not easily seen these labelled axons in the representative images provided in Figure 3A and 3C.

The authors should consider citing these papers which seem relevant to the idea tested (i.e. challenges and possibilities in treating chronic SCI) and to the discussion of the findings. PMID: 26134657, PMID: 11717367, PMID: 33975016

Reviewer #2: The introduction, rationale and stated hypotheses are the same as the approved Stage 1 Protocol submission and experiments were executed and analyzed as planned.

 I do have some questions about the presentation and conclusions presented in the abstract and discussion.

1. Overall there may be some tendency to advertise positive results but leave it to the reader to dig out the limitations and the effect size (which is quite small). The abstract in particular should give a more balanced and quantitative view of the data. The fact that RAGs were activated in DRG but not CST neurons is a significant finding. The fact that DRG and CST axon sprouting averaged less than 500 micros from the PROXIMAL edge of the lesion, and apparently didn't extend into tissue beyond the lesion, is certainly relevant information and informs the lack of behavioral effect. Bottom line, in my view the abstract should provide information about differences between the responses by different cell types and quantitative reference to the effect size of growth.

2. The injury is not sufficiently described, specifically the distinction between this injury, which is described as "severe" and a prior injury. This is important, because the difference in injury is offered as an explanation for the apparent reduction in DRG growth response in the prior acute study and the current chronic study.

From the present manuscript:

"A laminectomy at vertebra T9 was performed to expose spinal level T9 and a deep dorsal transection past the central canal leaving using micro-scissors (Fine Science Tools)."

From the prior manuscript (Joshi et al. 2015).

"A laminectomy at vertebra T9 was performed to expose spinal level T12 and a dorsal hemisection until the central canal was then performed using micro-scissors (Fine Science Tools)."

A more rigorous description of how depth was monitored, and how the depth differed between the two studies, is needed to support the claim that it can explain the reduced growth present here.

3. There seems to be a claim that TTK21 has different efficacy in sensory and motor neurons. 

"It is however important that TTK21 increases the growth and regenerative gene expression ability of both sensory and motor neurons, albeit with differential potency and efficacy."

Is this referring to a difference in the response of CST and DRG in the present data? I don't see that in the present data or a statistical test to support that claim.

4. "These findings are in line with previous work in subacute SCI where these classical RAGs were not activated in the corticospinal as opposed to DRG neurons[27]. "

The reference is Dr. Tuszynski's claim that CST neurons temporarily revert to an embryonic state, and doesn't seem to support the claim being made for differential gene activation in DRG versus CST neurons.

Reviewer #3: Muller and colleagues report their followup study regarding CBP/p300 activation in a model of delayed SCI (T9 partial (?) transection 1and treatment 12 weeks after injury). 

Abstract: The authors state: "The interruption of spinal circuitry following spinal cord injury disrupts neural activity AND IS FOLLOWED BY A FAILURE TO MOUNT AN EFFECTIVE REGENERATIVE GENE EXPRESSION RESPONSE resulting in permanent neurological disability." I believe that the statement in CAPS is not accurate. We reported in Nature 2020 that SCI DOES result in mounting regenerative gene expression, but the absence of a permissive milieu in the lesion site results in regenerative failure (Poplawski 2020; 581:77). I suggest that the authors simply state AND IS FOLLOWED BY A FAILURE TO MOUNT AN EFFECTIVE REGENERATIVE RESPONSE.

Related to the comment above, the authors may wish to reframe their conceptual context. We found as stated above that the regenerative state after SCI lasts for two weeks only. In a chronic state of injury (12-22 weeks after injury), this "primed" state for regeneration has closed. Thus, the authors may, by delivery of CSP-TTK, re-open this regenerative window. This concept is important from my perspective because we needs to identify means of re-opening the regenerative state.

Introduction: I suggest that you add PTEN and SOCS3 to the list genes that can influence regeneration, since the greatest evidence exists for PTEN. Although a phosphatase inhibitor, PTEN in effect acts like a transcription factor. 

Hypothesis statement: The authors state the following: "Here we hypothesize that the pharmacological stimulation of CBP/p300 activity will enhance regenerative gene expression during a growth refractory phase twelve weeks after spinal injury, while housing animals in an EE one-week post-injury will stimulate neuronal activity, consolidate axonal and synaptic plasticity as opposed to animals housed in SH." I find this somewhat confusing and disorienting. Is this a chronic study (12 weeks after injury) or a sub-acute study (EE one week after injury)? I find the entire Hypothesis section difficult to follow; I encourage the authors to re-write it to enhance clarity and simplicity.

Experimental Design: This section states that anatomical transection in a mouse allows more accurate anatomical definition of regeneration and sprouting than contusion in rat. Please simply state that anatomical transection allows more accurate anatomical assessment of regeneration and sprouting, since the choice of mice or rats has no bearing on transection vs. contusion. Also the authors suggest that T9 transection may spare some axons. This is not the case is the lesions are accurately performed. The lesion images shown in the figures appear to spare a substantial amount of spinal cord tissue. I think this point needs clarification: either it is advisable to omit reference to "severe" SCI, or to clarify why the images that show a substantial amount of sparing in Fig 3 are "severe".

Regarding the finding that TTK increased axonal growth but not functional recovery: Yes, TTK significantly increased axon growth, but the effect size was not very large. One might think that this is the reason that function did not recover. Perhaps this should be stated simply and clearly. If one goes from no regenerating axons to a few, this may not be likely to exert much of a benefit on function.

Hypothesize is misspelled in the paper ("Hypothesise"). 

I greatly appreciate that the authors are reporting the fact that these experimental manipulations do not improve functional outcomes. Too often in this field, these negative findings are omitted from a paper. This complete reporting enhances the qualify and credibility of the work.

---

## [Editor Report · Decision Letter 7]

12 Aug 2022

Dear Dr Di Giovanni,

Thank you for the submission of your revised Stage 2 Preregistered Research Article "CBP/p300 activation promotes axon growth, sprouting and synaptic plasticity in chronic experimental spinal cord injury with severe disability" for publication in PLOS Biology. On behalf of myself, my colleagues and the Academic Editors Cody Smith and Christopher Chambers, I am pleased to say that we can in principle accept your manuscript for publication, provided you address any remaining formatting and reporting issues. These will be detailed in an email you should receive within 2-3 business days from our colleagues in the journal operations team; no action is required from you until then. Please note that we will not be able to formally accept your manuscript and schedule it for publication until you have completed any requested changes.

PRESS

We frequently collaborate with press offices. If your institution or institutions have a press office, please notify them about your upcoming paper at this point, to enable them to help maximize its impact. If the press office is planning to promote your findings, we would be grateful if they could coordinate with biologypress@plos.org. If you have previously opted in to the early version process, we ask that you notify us immediately of any press plans so that we may opt out on your behalf.

Sincerely, 

Kris

Kris Dickson, Ph.D. (she/her)

Neurosciences Senior Editor/Section Manager

PLOS Biology

kdickson@plos.org